# Identifying gene expression programs of cell-type identity and cellular activity with single-cell RNA-Seq

Dylan Kotliar[1,2,3†]*, Adrian Veres[1,3,4†], M Aurel Nagy[3,5], Shervin Tabrizi[2], Eran Hodis[3,6], Douglas A Melton[4,7], Pardis C Sabeti[1,2,7]

[1]Department of Systems Biology, Harvard Medical School, Boston, United States; [2]Broad Institute of MIT and Harvard, Cambridge, United States; [3]Harvard-MIT Division of Health Sciences and Technology, Massachusetts Institute of Technology, Cambridge, United States; [4]Harvard Stem Cell Institute, Harvard University, Cambridge, United States; [5]Department of Neurobiology, Harvard Medical School, Boston, United States; [6]Biophysics Program, Harvard University, Cambridge, United States; [7]Howard Hughes Medical Institute, Chevy Chase, United States

**\*For correspondence:**
dylan_kotliar@hms.harvard.edu

[†]These authors contributed equally to this work

**Competing interests:** The authors declare that no competing interests exist.

**Abstract** Identifying gene expression programs underlying both cell-type identity and cellular activities (e.g. life-cycle processes, responses to environmental cues) is crucial for understanding the organization of cells and tissues. Although single-cell RNA-Seq (scRNA-Seq) can quantify transcripts in individual cells, each cell's expression profile may be a mixture of both types of programs, making them difficult to disentangle. Here, we benchmark and enhance the use of matrix factorization to solve this problem. We show with simulations that a method we call consensus non-negative matrix factorization (cNMF) accurately infers identity and activity programs, including their relative contributions in each cell. To illustrate the insights this approach enables, we apply it to published brain organoid and visual cortex scRNA-Seq datasets; cNMF refines cell types and identifies both expected (e.g. cell cycle and hypoxia) and novel activity programs, including programs that may underlie a neurosecretory phenotype and synaptogenesis.
DOI: https://doi.org/10.7554/eLife.43803.001

## Introduction

Genes act in concert to maintain a cell's identity as a specific cell type, to respond to external signals, and to carry out complex cellular activities such as replication and metabolism. Coordinating the necessary genes for these functions is frequently achieved through transcriptional co-regulation, where genes are induced together as a gene expression program (GEP) in response to the appropriate internal or external signal (*Eisen et al., 1998*; *Segal et al., 2003*). By enabling unbiased measurement of the whole transcriptome, profiling technologies such as RNA-Seq are paving the way for systematically discovering GEPs and shedding light on the biological mechanisms that they govern (*Liberzon et al., 2015*).

Single-cell RNA-Seq (scRNA-Seq) has greatly enhanced our potential to resolve GEPs by making it possible to observe variation in gene expression over many individual cells. Even so, inferring GEPs remains challenging as scRNA-Seq data is noisy and high-dimensional, requiring computational approaches to uncover the underlying patterns. In addition, technical artifacts such as doublets (where two or more distinct cells are mistakenly collapsed into one) can confound analysis. Methodological advances in dimensionality reduction, clustering, lineage trajectory tracing, and differential expression analysis have helped overcome some of these issues (*Amir et al., 2013*; *Kharchenko et al., 2014*; *Satija et al., 2015*; *Trapnell et al., 2014*).

Here, we focus on a key challenge of inferring expression programs from scRNA-Seq data: the fact that individual cells may express multiple GEPs but we only detect cellular expression profiles that reflect their combination, rather than the GEPs themselves. A cell's gene expression is shaped by many factors including its cell type, its state in time-dependent processes such as the cell cycle, and its response to varied environmental stimuli (*Wagner et al., 2016*). We group these into two broad classes of expression programs that can be detectable in scRNA-Seq data: (1) GEPs that correspond to the identity of a specific cell type such as hepatocyte or melanocyte (identity programs) and (2) GEPs that are expressed independently of cell type, in any cell that is carrying out a specific activity such as cell division or immune cell activation (activity programs). In this formulation, identity programs are expressed uniquely in cells of a specific cell type, while activity programs may vary dynamically in cells of one or multiple types and may be continuous or discrete.

Thus far, the vast majority of scRNA-Seq studies have focused on systematically identifying and characterizing the expression programs of cell types composing a given tissue, that is identity GEPs. Substantially less progress has been made in identifying activity GEPs, primarily through direct manipulation of cells in controlled experiments, for example comparing stimulated and unstimulated neurons (*Hrvatin et al., 2018*) or cells pre- and post-viral infection (*Steuerman et al., 2018*).

If a subset of cells profiled by scRNA-Seq expresses a given activity GEP, there is a potential to directly infer the program from the data without the need for controlled experiments. However, this can be significantly more challenging than ascertaining identity GEPs; while some cells may have expression profiles that are predominantly the output of an identity program, activity programs will always be expressed alongside the identity programs of one or frequently many cell types. Thus, while finding the average expression of clusters of similar cells may often be sufficient for finding reasonably accurate identity GEPs, it will often fail for activity GEPs.

We hypothesized that we could infer activity GEPs directly from variation in single-cell expression profiles using matrix factorization. In this context, matrix factorization would model the gene expression data matrix as the product of two lower rank matrices, one encoding the relative contribution of each gene to each program, and a second specifying the proportions in which the programs are combined for each cell. We refer to the second matrix as a 'usage' matrix as it specifies how much each GEP is 'used' by each cell in the dataset (*Stein-O'Brien et al., 2018*) (*Figure 1A*). Unlike hard clustering, which reduces all cells in a cluster to a single shared GEP, matrix factorization allows cells to express multiple GEPs. Thus, this computational approach would allow cells to express one or more activity GEPs in addition to their expected cell-type GEP, and could correctly model doublets as a combination of the identity GEPs for the combined cell types. To the best of our knowledge, no previously reported studies have benchmarked the ability of matrix factorization methods to accurately learn identity and activity GEPs from scRNA-Seq profiles.

We see three primary motivations for jointly inferring identity and activity GEPs in scRNA-Seq data. First, systematic discovery of GEPs could reveal unexpected or novel activity programs reflecting important biological processes (e.g. immune activation or hypoxia) in the context of the native biological tissue. Second, it could enable characterization of the prevalence of each activity GEP across cell types in the tissue. Finally, accounting for activity programs could improve inference of identity programs by avoiding spurious inclusion of activity program genes in the latter. GEPs corresponding to different phases of the cell cycle are examples of widespread activity programs and are well-known to confound identity (cell type) program inference in scRNA-Seq data (*Chen and Zhou, 2017*; *Scialdone et al., 2015*). However, cell-cycle is just one instance of the broader problem of confounding of identity and activity programs.

While matrix factorization is widely used as a preprocessing step in scRNA-Seq analysis, a priori it is unclear which, if any, factorization approaches would be most appropriate for inferring biologically meaningful GEPs. In particular, Principal Component Analysis (PCA), Independent Component Analysis (ICA), Latent Dirichlet Allocation (LDA) (*Blei et al., 2003*) and Non-Negative Matrix Factorization (NMF)(*Lee and Seung, 1999*) have been used for dimensionality reduction of data prior to downstream analysis or as an approach to cell clustering. However, while PCA (*Shalek et al., 2014*; *Steuerman et al., 2018*), NMF (*Puram et al., 2017*) and ICA (*Saunders et al., 2018*) components have been interpreted as activity programs, the dimensions inferred by these or other matrix factorization algorithms may not necessarily align with biologically meaningful gene expression programs and are frequently ignored in practice. This is because each method makes different simplifying assumptions that are potentially inappropriate for gene expression data. For example, NMF and

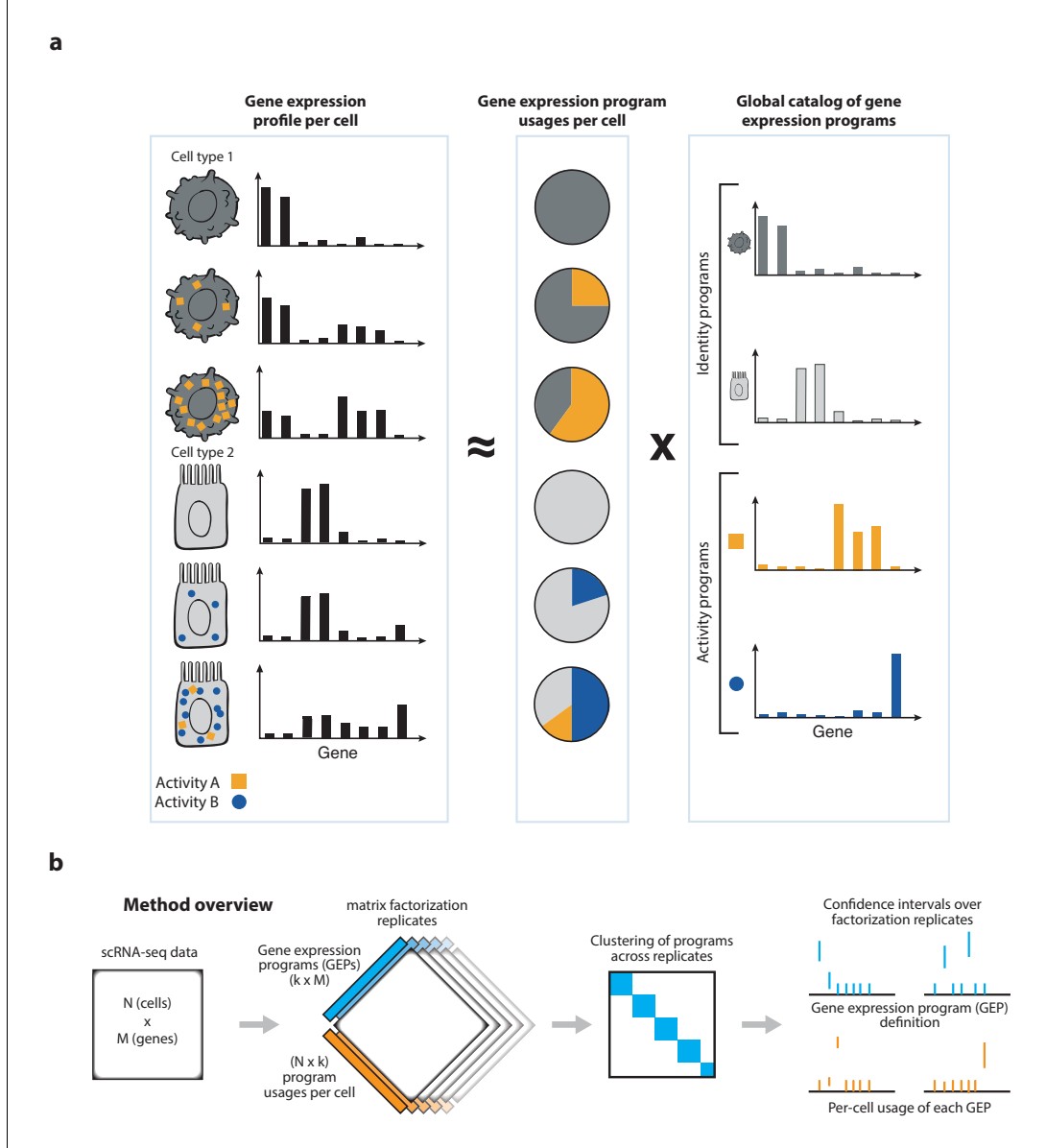

**Figure 1.** Schematics of matrix factorization for single-cell RNA-Seq analysis and the consensus matrix factorization pipeline. (**a**) Schematic representation of how cellular gene expression can be modeled with matrix factorization. The gene expression profiles of individual cells (left bar charts) are weighted mixtures of a set of global gene expression programs (right bar charts) with distinct weights reflecting the usage of each program (middle pie chart). Variable usage of the activity programs is represented by the number of blue circles and orange squares in each cell. (**b**) Schematic of the consensus matrix factorization pipeline.

DOI: https://doi.org/10.7554/eLife.43803.002

LDA are non-negative and so cannot directly model repression. ICA components are statistically independent, PCA components are mutually orthogonal, and both allow gene expression to be negative. Furthermore, none of these methods, except LDA, explicitly accounts for the count distribution of expression data in their error models.

In this study, we motivate, validate, and enhance the use of matrix factorization for GEP inference. Using simulations, we show that despite their simplifying assumptions, ICA, LDA, and NMF—but not PCA—can accurately discover both activity and identity GEPs. However, due to inherent randomness in their algorithms, they give substantially varying results when repeated multiple times, which hinders their interpretability. We therefore implemented a meta-analysis approach (*Figure 1B*), which

demonstrably increased robustness and accuracy. Overall, the meta-analysis of NMF, which we call Consensus NMF (cNMF), gave the best performance in these simulations.

Applied to three real datasets generated by three different scRNA-Seq platforms, cNMF inferred expected activity programs (cell-cycle programs in a brain organoid dataset and depolarization induced programs in visual cortex neurons), an unanticipated hypoxia program, and intriguing novel activity programs. It also enhanced cell type characterization and enabled estimation of rates of activity across cell types. These findings on real datasets further validate our approach as a useful analysis tool to understand complex signals within scRNA-Seq data.

## Results

### Evaluation of matrix factorization for GEP inference in simulated data

We sought to establish whether components inferred by simple matrix factorizations would align with GEPs in scRNA-seq data. We evaluated this in simulated data of 15,000 cells composed of 13 cell types, one cellular activity program that is active to varying extents in a subset of cells of four cell types, and a 6% doublet rate (*Figure 2A*). We generated 20 replicates of this simulation, each at three different 'signal to noise' ratios, in order to determine how matrix factorization accuracy varies with noise level (Materials and methods).

We first analyzed the performance of ICA, LDA, and NMF and noticed that they yielded different solutions when run several times on the same input simulated data. We ran each method 200 times and assigned the components in each run to their most correlated ground-truth program. We saw that there was significant variability among the components assigned to the same program – particularly for NMF and LDA (*Figure 2—figure supplement 1*). Unlike PCA, which has an exact solution, these factorizations use stochastic optimization algorithms to obtain approximate solutions in a solution space including many local optima. We observed that such local optima frequently corresponded to solutions where a simulated GEP was split into multiple inferred components and/or multiple GEPs were merged into a single component (*Figure 2—figure supplement 2a*). This variability reduces the interpretability of the solutions and may decrease the accuracy as well.

To overcome the issue of variability of solutions, we employed a meta-analysis approach, which we call consensus matrix factorization, that averages over multiple replicates to increase the robustness of the solution. The method which is adapted from a similar procedure in mutational signature discovery (*Alexandrov et al., 2013*) proceeds as follows: we run the factorization multiple times, filter outlier components (which tend to represent noise or merges/splits of GEPs), cluster the components over all replicates combined, and take the cluster medians as our consensus estimates. With these estimates fixed, we are able to compute a final usage matrix specifying the contribution of each GEP in each cell and to transform our GEP estimates from normalized units to biologically meaningful ones such as transcripts per million (TPM). This approach also provides us with a guide for determining K, the number of components to use, by selecting a value that provides a reasonable trade-off between error and stability (*Figure 2—figure supplement 3a*, see Materials and methods for details). We refer to this approach as consensus matrix factorization based on its analogy with consensus clustering (*Monti et al., 2003*) and to its application to LDA, NMF, and ICA, as cLDA, cNMF, and cICA respectively. While consensus clustering has been previously applied to bulk gene expression analysis using hard-clustering derived by binarizing NMF factors (*Brunet et al., 2004*), our approach does not require any hard cluster assignments.

Consensus matrix factorization inferred components underlying the GEPs as well as which cells expressed each GEP (*Figure 2b–c*, *Figure 2—figure supplement 4a*). By contrast, principal components were linear combinations of the true GEPs. Beyond increasing the robustness of the solution, the consensus approach also increased the ability of factorization to deconvolute the true GEPs - most dramatically for LDA and NMF which had the most stochastic variability. cNMF successfully deconvoluted the activity and identity GEPs more frequently than the other matrix factorizations considered (*Figure 2d*, *Figure 2—figure supplement 2*).

We next sought to benchmark the sensitivity and specificity of each matrix factorization method for inferring which genes are associated with each GEP. We also evaluated the performance of hard clustering for this task because clustering is the most common way GEPs are identified in practice. We evaluated the commonly used Louvain community detection clustering algorithm (*Blondel et al.,*

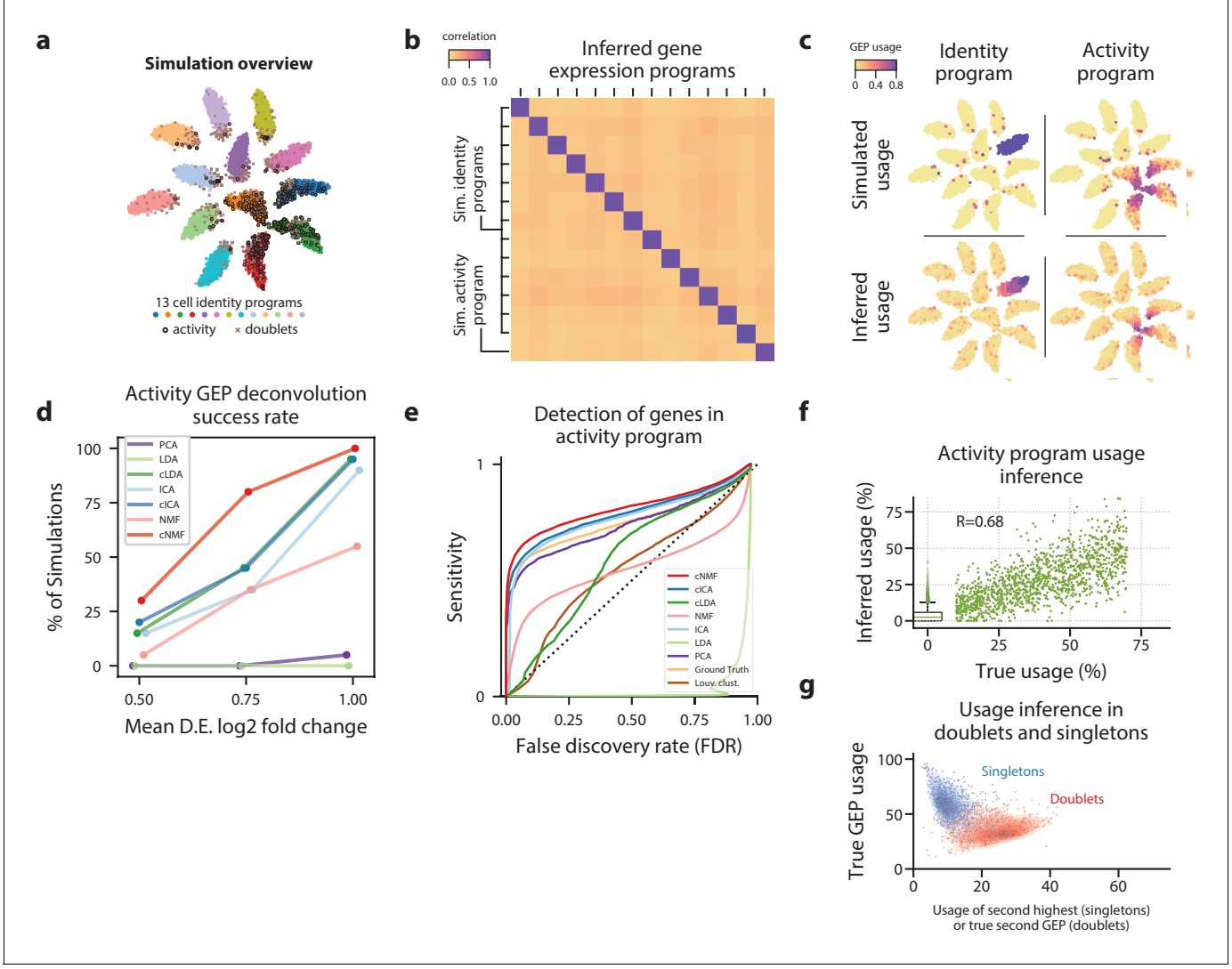

**Figure 2.** cNMF infers identity and activity expression programs in simulated data. (**a**) t-distributed stochastic neighbor embedding (tSNE) plot of an example simulation showing different cell types with marker colors, doublets as gray Xs, and cells expressing the activity gene expression program (GEP) with a black edge. (**b**) Pearson correlation between the true GEPs and the GEPs inferred by cNMF for the simulation in (**a**). (**c**) Same tSNE plot as (**a**) but colored by the simulated or the cNMF inferred usage of an example identity program (left) or the activity program (right). (**d**) Percentage of 20 simulation replicates where an inferred GEP had Pearson correlation greater than 0.80 with the true activity program for each signal to noise ratio (parameterized by the mean log2 fold-change for a differentially expressed gene). (**e**) Receiver Operator Characteristic (except with false discovery rate rather than false positive rate) showing prediction accuracy of genes associated with the activity GEP. (**f**) Scatter plot comparing the simulated activity GEP usage and the usage inferred by cNMF for the simulation in (**a**). For cells with a simulated usage of 0, the inferred usage is shown as a box and whisker plot with the box corresponding to interquartile range and the whiskers corresponding to 5th and 95th percentiles. (**g**) Contour plot of the true GEP usage on the Y-axis and the second true GEP usage for doublets or the second highest GEP usage inferred by cNMF for singletons for the simulation in (**a**). 1000 randomly selected cells are overlayed as a scatter plot for each group.

DOI: https://doi.org/10.7554/eLife.43803.003

The following source data and figure supplements are available for figure 2:

**Source data 1.** Benchmarking of matrix factorizations on simulated data.
DOI: https://doi.org/10.7554/eLife.43803.012
**Figure supplement 1.** Robustness of matrix factorization methods.
DOI: https://doi.org/10.7554/eLife.43803.004
**Figure supplement 2.** Deconvolution accuracy of matrix factorization methods.
DOI: https://doi.org/10.7554/eLife.43803.005
**Figure supplement 3.** Diagnostic plot for cNMF on an example simulated dataset.
DOI: https://doi.org/10.7554/eLife.43803.006

*Figure 2 continued on next page*

*Figure 2 continued*

**Figure supplement 4.** GEP usage inference.
DOI: https://doi.org/10.7554/eLife.43803.007
**Figure supplement 5.** Accuracy of identifying genes in each GEP.
DOI: https://doi.org/10.7554/eLife.43803.008
**Figure supplement 6.** Comparison of run-times for different matrix factorization algorithms.
DOI: https://doi.org/10.7554/eLife.43803.009
**Figure supplement 7.** cNMF demonstration on simulated datasets with 50% doublets.
DOI: https://doi.org/10.7554/eLife.43803.010
**Figure supplement 8.** Impact of variable cell-type proportions on GEP inference.
DOI: https://doi.org/10.7554/eLife.43803.011

*2008*; *Levine et al., 2015*) but also considered an upper bound on how well any discrete clustering could perform by using ground-truth to assign cells to a cluster of its cell type or to an activity cluster if it had >= 40% simulated contribution from the activity GEP (*Figure 2—figure supplement 4b*). We evaluated the association between genes and GEPs using linear regression and measured accuracy using a receiver operator characteristic (Materials and methods).

We found that cNMF was most accurate at inferring genes in the activity program, with a sensitivity of 61% at a false discovery rate (FDR) of 5% (*Figure 2e*). cICA and the ground-truth clustering were the next most accurate with 57% and 56% sensitivity at a 5% FDR, respectively. cNMF also performed the best at inferring identity GEPs of the 4 cell types that expressed the activity (*Figure 2—figure supplement 5*). As expected, the clustering approaches performed worse as they inappropriately assigned activity GEP genes to these identity programs, resulting in an elevated FDR. This illustrates how matrix factorization can outperform clustering for inference of the genes associated with activity and identity GEPs.

We decided to proceed with cNMF to analyze the real datasets due its accuracy, processing speed, and interpretability. First, it yielded the most accurate inferences in our simulated data. Second, NMF was the fastest of the basic factorization algorithms considered, which is especially useful given the need to run multiple replicates and given the growing sizes of scRNA-Seq datasets (*Figure 2—figure supplement 6*). Third, the non-negativity assumption of NMF naturally results in usage and component matrices that can be normalized and interpreted as probability distributions— that is, where the usage matrix reflects the probability of each GEP being used in each cell, and the component matrix reflects the probability of a specific transcript expressed in a GEP being a specific gene. The other high-performing factorization method, cICA, produced negative values in the components and usages which precludes this interpretation.

Beyond identifying the activity program itself, we found that cNMF could also accurately infer which cells expressed the activity program and what proportion of their expression was derived from the activity program (*Figure 2f*). With an expression usage threshold of 10%, cNMF accurately classified 91% of cells expressing the activity program and 94% of cells that did not express the program. Moreover, we observed a high Pearson correlation between the inferred and simulated usages in cells that expressed the program (R = 0.74 for all simulations combined, R = 0.68 for the example simulation in *Figure 2a*). Thus, cNMF can be used both to infer which cells express the activity program, as well as what proportion of their transcripts derive from that program.

We further demonstrated that cNMF was robust to the presence of doublets—instances where two cells are mistakenly labeled as a single cell. Due to limitations in the current tissue dissociation and single-cell sequencing technologies, some number of 'cells' in an scRNA-Seq dataset will actually correspond to doublets. Several computational methods have been developed to identify cells that correspond to doublets, but this is still an important artifact in scRNA-Seq data (*McGinnis et al., 2018*; *Wolock et al., 2018*). We found that cNMF correctly modeled doublets as a combination of the GEPs for the two combined cell types (*Figure 2g*). Moreover, we found that cNMF could accurately infer the GEPs even in a simulated dataset composed of 50% doublets (*Figure 2—figure supplement 7*). This illustrates another benefit of representing cells in scRNA-Seq data as a mixture of GEPs rather than classifying them into discrete clusters.

In all the simulations described above, the 13 cell-types occurred at uniform frequencies. This allowed us to treat all identity programs as replicates of each other for evaluating inference

accuracy, rather than having to separately consider rare GEPs which should, all else equal, be harder to infer than common ones. However, this is an approximation of reality where cell-type proportions can vary over multiple orders of magnitude. We therefore also performed simulations containing biologically plausible cell-type proportions derived from the published clustering of a dataset analyzed later in this manuscript (*Hrvatin et al., 2018*) (Materials and methods). When we kept all of the other simulation parameters identical to those of the initial simulations, some identity GEPs from rare cell-types were missed by cNMF, cICA, and Louvain clustering (*Figure 2—figure supplement 8a*). However, when we increased the distinctness of the identity GEPs of the cell types, they could still be inferred by both cICA and cNMF with similar relative performances to what we saw in the primary benchmarking analysis (*Figure 2—figure supplement 8b*). This suggests that the simplification of uniform cell-type frequencies does not significantly impact our conclusions.

## cNMF deconvolutes hypoxia and cell-cycle activity GEPs from identity GEPs in brain organoid data

Having demonstrated its performance and utility on simulated data, we then used cNMF to re-analyze a published scRNA-Seq dataset of 52,600 single cells isolated from human brain organoids (*Quadrato et al., 2017*). The initial report of this data confirmed that organoids contain excitatory cell types homologous to those in the cerebral cortex and retina as well as unexpected cells of mesodermal lineage, but further resolution can be gained on the precise cell types and how they differentiate over time. As organoids contain many proliferating cell types, we sought to use this data to confirm that cNMF could detect activity programs—in this case, cell cycles programs—in real data, and to explore what biological insights could be gained from their identification.

We identified 31 distinct programs in this dataset that could be further parsed into identity and activity programs (*Figure 3—figure supplement 1*). We distinguished between identity and activity programs by using the fact that activity programs can occur in multiple diverse cell types while identity programs represent a single-cell type. Most cells had high usage of just a single GEP, which is consistent with expressing just an identity program (*Figure 3a*). When cells expressed multiple GEPs, those typically had correlated expression profiles, suggesting that they correspond to identity programs of closely related cell types or cells transitioning between two developmental states, rather than activity programs (*Figure 3—figure supplement 2*). By contrast, three GEPs were co-expressed with many distinct and uncorrelated programs, suggesting that they represent activity programs that occur across diverse cell types (*Figure 3a–b*). Consistent with this, the 28 suspected identity programs were well separated by the cell-type clusters reported in *Quadrato et al. (2017)* while the three suspected activity programs were expressed by cells across multiple clusters (*Figure 3—figure supplements 3–4*). Except for a few specific cases discussed below, we used these published cluster labels to annotate our identity GEPs.

Our 28 identity programs further refined the 10 primary cell-type clusters originally reported for this dataset. For example, we noticed that cells previously annotated as mesodermal predominantly expressed one of three GEPs that were significantly enriched for genes in the 'Muscle Contraction' Gene Ontology (GO) set ($p < 1 \times 10^{-10}$ vs. $p > 0.19$ for all other GEPs). They therefore likely represent muscle cells. Inspecting the genes associated with these three GEPs, we noticed that they include genes characteristic of different classes of skeletal muscle: (1) immature skeletal muscle (e.g. *MYOG*, *TNNT2*, *NES*), (2) fast-twitch muscle (e.g. *TNNT3*, *TNNC2*, *MYOZ1*), and (3) slow-twitch muscle (e.g. *TNNT1*, *TNNC1*, *TPM3*) (*Figure 3d*, *Supplementary file 1*). This unexpected finding suggests that distinct populations of skeletal muscle cells – excitatory cell types with many similarities to neurons – are differentiating in these brain organoids.

Of the three activity programs identified, we found that two were strongly enriched for cell cycle Gene Ontology (GO) sets, suggesting that they correspond to separate phases of the cell cycle (*Figure 3c*). One showed stronger enrichment for genesets involved in DNA replication (e.g. DNA Replication $p = 3 \times 10^{-52}$ compared to $p = 2 \times 10^{-3}$) while the other showed stronger enrichment for genesets involved in mitosis (e.g. Mitotic Nuclear Division, $p = 4 \times 10^{-61}$ compared to $p = 2 \times 10^{-46}$). These enrichments and inspection of the genes most associated with these programs implied that one represents a G1/S checkpoint program and the other represents a G2/M checkpoint program (*Figure 3e*). Thus, cNMF discovered two activity programs corresponding to separate phases of the cell cycle directly from the data.

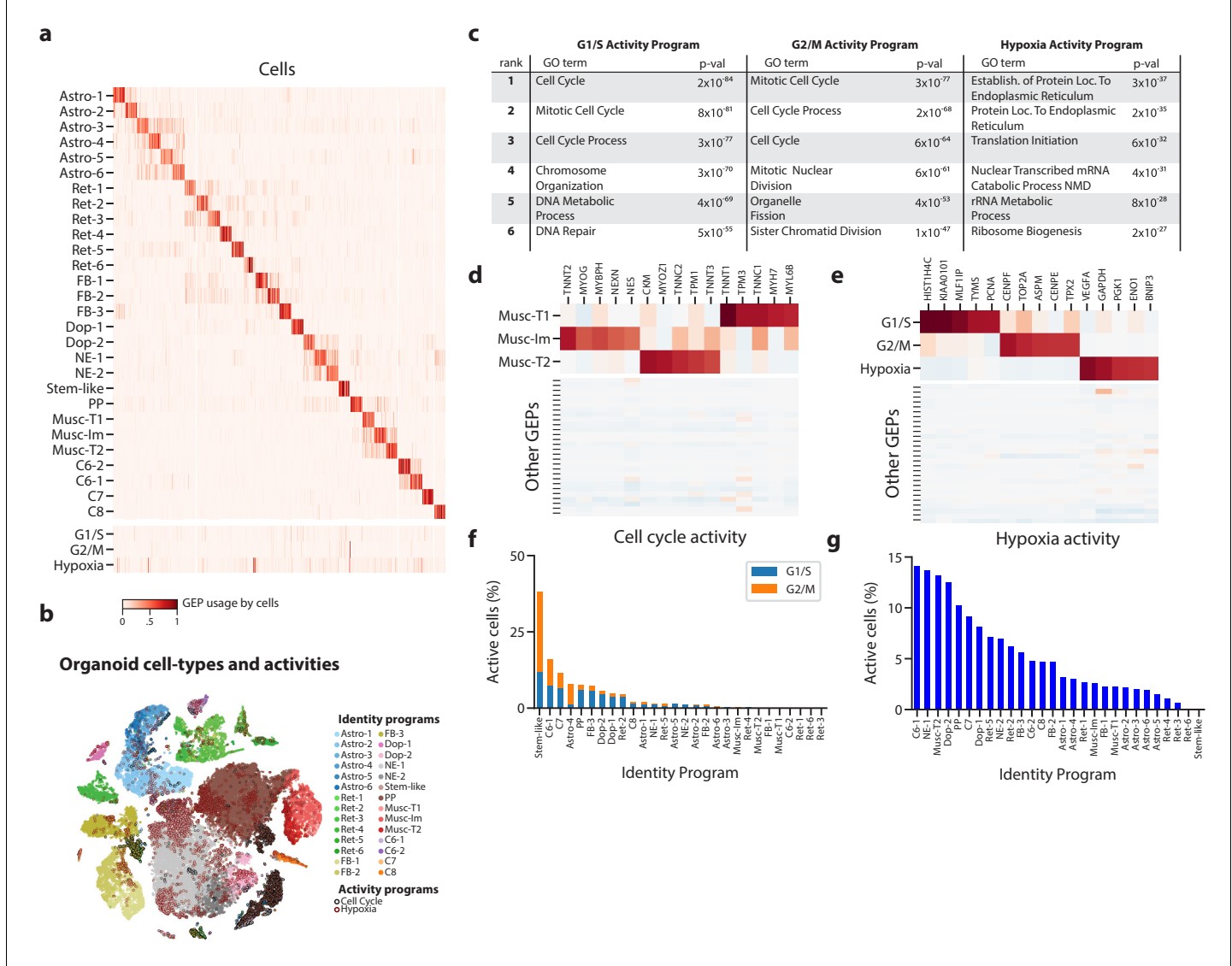

**Figure 3.** Deconvolution of activity programs from cell identity in brain organoid data. (**a**) Heatmap showing percent usage of all GEPs (rows) in all cells (columns). Identity GEPs are shown on top and activity GEPs are shown below. Cells are grouped by their maximum identity GEP and fit into columns of a fixed width for each identity GEP. (**b**) tSNE plot of the brain organoid dataset with cells colored by their maximally used identity GEP, and with a black edge for cells with >10% usage of the G1/S or G2/M activity GEP or a maroon edge for cells with >10% usage of the hypoxia GEP. (**c**) Table of P-values for the top six Gene Ontology geneset enrichments for the three activity GEPs. (**d**) Heatmap of z-scores of top genes associated with three mesodermal programs, in those programs (top), and in all other programs (bottom). (**e**) Heatmap of z-scores of top genes associated with three activity GEPs, in those programs (top), and in all other programs (bottom). (**f**) Proportion of cells assigned to each identity GEP that express the G1/S or G2/M program with a percent usage greater than 10%. (**g**) Proportion of cells assigned to each identity GEP that express the hypoxia program with a percent usage greater than 10%.

DOI: https://doi.org/10.7554/eLife.43803.013

The following source data and figure supplements are available for figure 3:

**Source data 1.** Application of cNMF to brain organoid data.
DOI: https://doi.org/10.7554/eLife.43803.018

**Figure supplement 1.** Diagnostic plot for cNMF on the *Quadrato et al. (2017)* brain organoid dataset.
DOI: https://doi.org/10.7554/eLife.43803.014

**Figure supplement 2.** Correlation between GEP spectra pairs and fraction of cells that use both programs.
DOI: https://doi.org/10.7554/eLife.43803.015

**Figure supplement 3.** t-SNE plots of identity and activity GEPs in the *Quadrato et al. (2017)* brain organoid dataset.
DOI: https://doi.org/10.7554/eLife.43803.016

*Figure 3 continued on next page*

*Figure 3 continued*

**Figure supplement 4.** Comparison of cNMF usages with the cell-type clusters from *Quadrato et al. (2017)*.
DOI: https://doi.org/10.7554/eLife.43803.017

The third activity program is characterized by high levels of hypoxia related genes (e.g. *VEGFA*, *PGK1*, *CA9*, *P4HA1*, *HILPDA*) suggesting it represents a hypoxia program (*Figure 3e*). This is consistent with the lack of vasculature in organoids which makes hypoxia an important growth constraint (*Kelava and Lancaster, 2016*). This GEP is most significantly enriched for genesets related to protein localization to the endoplasmic reticulum and nonsense mediated decay ($p=3\times10^{-37}$, $p=5\times10^{-31}$) (*Figure 3c*), consistent with reports that hypoxia post-transcriptionally increases expression of genes that are translated in the ER (*Staudacher et al., 2015*) and modulates nonsense mediated decay activity (*Gardner, 2008*). In the initial report of this data, staining for a single hypoxia gene, *HIF1A*, failed to detect significant levels of hypoxia. Indeed, *HIF1A* is not strongly associated with this GEP, at least not at the transcriptional level. This highlights the ability of our unbiased approach to detect unanticipated activity programs in scRNA-Seq data.

Having identified proliferation and hypoxia activity programs, we sought to quantify their relative rates across cell types in the data. We found that 3079 cells (5.9%) expressed the G1/S program and 2043 cells (3.9%) expressed the G2/M program (with usage >= 10%). Classifying cells into cell types according to their most used identity program, we found that many distinct populations were replicating. For example, cNMF detected a rare population, included with the forebrain cluster in the original report, that we label as 'stem-like' based on high expression of pluripotency markers (e.g. *LIN28A*, *L1TD1*, *MIR302B*, *DNMT3B*) (*Supplementary file 1*). These cells showed the highest rates of proliferation with over 38% of them expressing a cell-cycle program in addition to the 'stem-like' identity GEP (*Figure 3f*).

cNMF was further able to refine cell types by disentangling the contributions of identity and activity programs to the gene expression of cells. For example, we found that a cell cluster labeled in *Quadrato et al. (2017)* as 'proliferative precursors', based on high expression of cell-cycle genes, is composed of multiple cell types including immature muscle and dopaminergic neurons (*Figure 3—figure supplement 4*). The predominant identity GEP of cells in this cluster is most strongly associated with the gene PAX7, a marker of self-renewing muscle stem cells (*Pawlikowski et al., 2009*) (*Supplementary file 1*). Indeed, this GEP has high (>10%) usage in 41% of cells who's most used GEP is the immature muscle program, suggesting it may be a precursor of muscle cells. This relationship was not readily identifiable by clustering because the majority of genes associated with the cluster were cell cycle related.

We also saw a wide range of cell types expressing the hypoxia program, with the highest rates in C6-1, neuroepithelial-1, type 2 muscle, and dopaminergic-2 cell types. The lowest levels of hypoxia program usage occurred in forebrain, astroglial, retinal, and type 1 muscle cell types (*Figure 3g*). The hypoxia response program is widespread in this dataset with 5788 cells (11%) of all cells expressing it (usage >10%). This illustrates how inferring activity programs in scRNA-Seq data using cNMF makes it possible to compare the rates of cellular activities across cell types.

## cNMF identifies depolarization induced and novel activity programs in scRNA-Seq of mouse visual cortex neurons

Next we turned to another published dataset to further validate cNMF and to illustrate how it can be combined with scRNA-Seq of experimentally manipulated cells to uncover more subtle activity programs. We re-analyzed scRNA-Seq data from 15,011 excitatory pyramidal neurons or inhibitory interneurons from the visual cortex of dark-reared mice that were suddenly exposed to 0 hr, 1 hr, or 4 hr of light (*Hrvatin et al., 2018*). This allowed the authors to identify transcriptional changes induced by repeated depolarization, a phenomenon believed to be critical for proper cortical function. To increase our resolution to identify neuronal activity programs, we used the published clustering to exclude any non-neuronal cells that were also captured by the experiment from our dataset. We sought to determine whether cNMF would identify the relatively modest activity programs (~60 genes with fold-change >= 2 and FDR < 0.05) elicited by the experiment without knowledge of the experimental design labels. Furthermore, since the authors identified heterogeneity in stimulus-

responsive genes between neuronal subtypes, we wondered if cNMF would identify a common activity program and whether it could tease out patterns in what is shared or divergent across neuron subtypes.

We ran cNMF on neurons combined from all three exposure conditions and identified 20 GEPs, interpreting 14 as identity and six as activity programs (*Figure 4—figure supplement 1*). As we saw in the organoid data, the activity programs were co-expressed with many distinct and uncorrelated GEPs while the identity programs only overlapped in related cell types (*Figure 4a–b*). In addition, the identity programs were well separated by the published clusters while the activity programs were spread across multiple clusters (*Figure 4—figure supplement 2*). We thus used the published cluster annotations to label the identity GEPs.

Three activity programs were correlated with the stimulus, which indicates that they are induced by repeated depolarization (*Figure 4c*). One of these was induced at 1h and thus corresponds to an early response program (ERP). The others were primarily induced at 4h and thus correspond to late response programs (LRPs). These programs overlapped significantly with the sets of differentially expressed genes reported in *Hrvatin et al. (2018)* (p=$8\times10^{-34}$ for the ERP and genes induced at 1h; p=$4\times10^{-22}$, p=$4\times10^{-14}$ for the LRPs and genes induced at 4hs, one-sided Mann Whitney test).

Intriguingly, one LRP was more induced in superficial cortical layers, while the other was more induced in deeper layers. This supports a recently proposed model where the ERP is predominantly shared across excitatory neurons, while LRPs vary more substantially across neuron subtypes (*Hrvatin et al., 2018*). It also illustrates cNMF's sensitivity: in the initial report, only 64 and 53 genes were identified as differentially expressed in at least one excitatory cell type at 1h and 4hs (FC $\geq$2, FDR < 0.05). Nevertheless, cNMF was able to find this program in an unsupervised manner, without knowledge of the experimental design. cNMF was also able to identify a depolarization induced program in visual cortex neurons that were not experimentally manipulated to elicit them. We analyzed an additional scRNA-Seq dataset of 1573 neurons from the visual cortex of adult mice that, unlike in the primary dataset, were not reared in darkness or treated with a specific light stimulus (*Tasic et al., 2016*). In this dataset, cNMF identified a matching GEP for all visual cortex cell types found in the primary dataset except for a single subtype of excitatory layer 5 (*Figure 4—figure supplement 3a*). Moreover, it identified a GEP that showed striking concordance with the superficial LRP found in the primary dataset (Fisher Exact Test of genes with association z-score >0.0015, OR = 127, p=$1\times10^{-118}$, Pearson Correlation = 0.645) (*Figure 4d*). This program was predominantly expressed in excitatory cells of the more superficial layers of the cortex as would be expected based on the results in the primary dataset. For example, over 40% of the excitatory layer 2 (Exc. L2) type neurons expressed this activity program (*Figure 4—figure supplement 3b*). This demonstrates that cNMF could also find the depolarization induced activity program in scRNA-Seq of cells that had not been experimentally manipulated to elicit it.

Finally, cNMF identified three additional activity programs in the primary visual cortex dataset that were not well correlated with the light stimulus but were expressed broadly across multiple neuronal cell types (*Figure 4b–c*). We labeled one of these, that was specific to excitatory neurons, as Neurosecretory (NS) because it is characterized by high expression of several secreted neuropeptides including *Vgf, Adcyap1, Scg2, Cck, Scg3,* and *Dkk3,* and has high expression of genes that facilitate protein secretion such as *Cpe, Cadps2,* and *Scamp5* (*Supplementary file 2*). The top expressed gene—*Vgf* (VGF nerve growth factor inducible) is induced by nerve growth factor (*Salton et al., 1991*), suggesting that this program may be regulated by external growth factor signals. Notably, we found a matching program in the Tasic Et, Al. dataset (Fisher Exact Test of genes with association z-score >0.0015, OR = 53.8, p=$8\times10^{-21}$, Pearson Correlation = 0.356) (*Figure 4e*). Thus, this neurosecretory activity program is reproducible across multiple single-cell datasets.

An additional activity program which we labeled Synaptogenesis (Syn) was characterized by expression of genes that play a crucial role in synapse formation, including the transcriptional regulator *Mef2c* (*Barbosa et al., 2008*; *Flavell et al., 2008*; *Harrington et al., 2016*), synaptic adhesion molecules *Ncam1* (*Hata et al., 2018*) and *Cadm1* (*Biederer et al., 2002*; *Robbins et al., 2010*), membrane vesicle traffickers *Syt1* and *Syt4, Actb* which constitutes the predominant synapse cytoskeletal protein, and others with a strong connection to synapse biology such as *Ywhaz* (*Foote et al., 2015*; *Ramser et al., 2010*; *Xu et al., 2015*), *Bicd1* (*Aguirre-Chen et al., 2011*; *Li et al., 2010*). It was also significantly enriched for relevant Gene Ontology sets including postsynapse, glutamatergic synapse, postsynaptic density, and dendrite morphogenesis (p<=$3.25\times10^{-6}$,

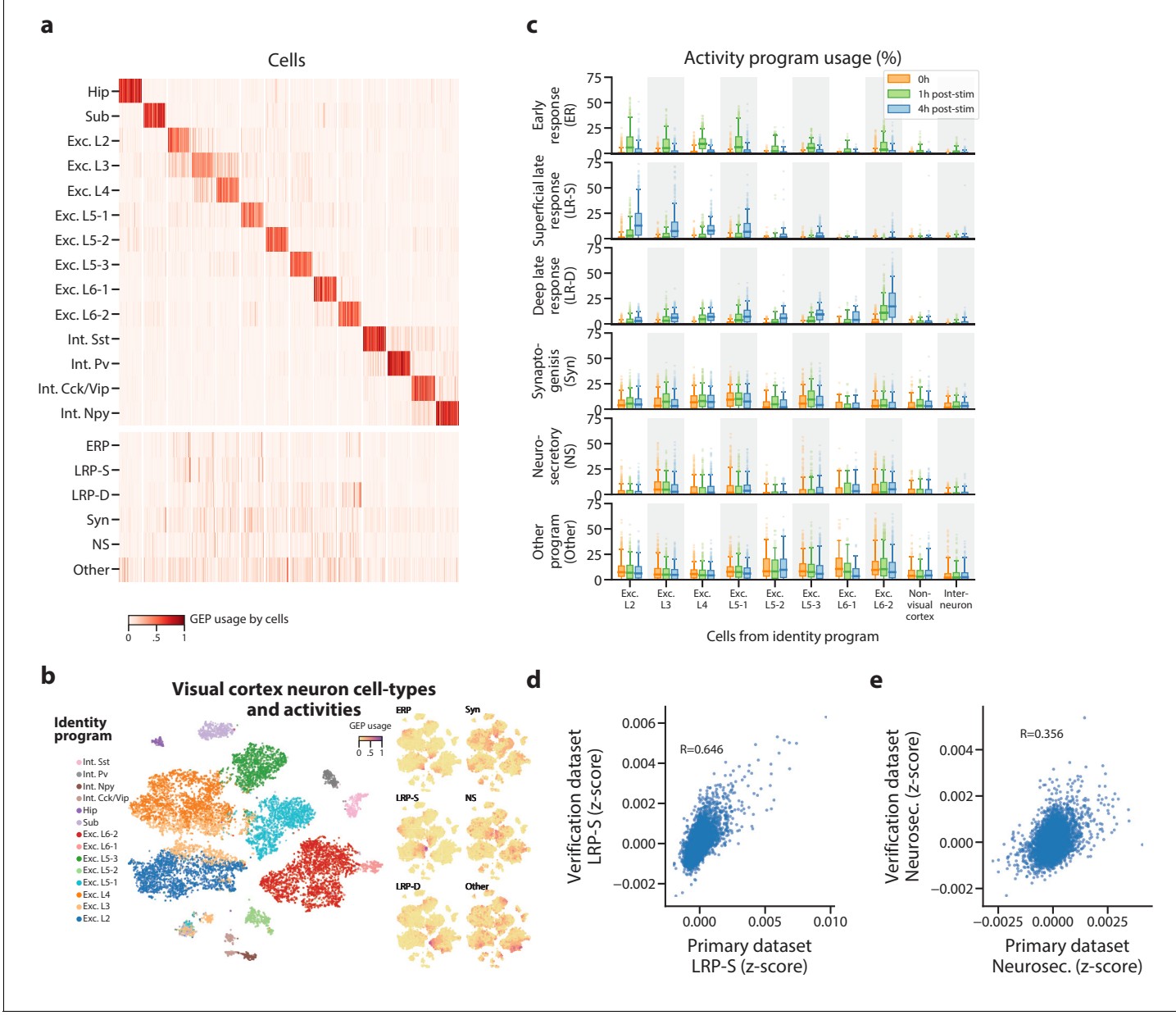

**Figure 4.** Identification of activity programs in neurons of the visual cortex. (**a**) Heatmap showing percent usage of all GEPs (rows) in all cells (columns). Identity GEPs are shown on top and activity GEPs are shown below. Cells are grouped by their maximum identity GEP and fit into columns of a fixed width for each identity GEP. (**b**) t-SNE plots of cells colored by maximum identity GEP usage (left) or by absolute usage of each activity GEP (right). (**c**) Box and whisker plot showing the percent usage of activity programs (rows) in cells classified according to their maximum identity GEP (columns) and stratified by the stimulus condition of the cells (hue). The central line represents the median, boxes represent the interquartile range, and whiskers represent the 5th and 95th quantile. (**d**) Scatter plot of z-scores of the superficial late response GEP in the primary dataset against the corresponding GEP in the *Tasic et al. (2016)* dataset. (**e**) Same as (**d**) but for the neurosecretory activity program.

DOI: https://doi.org/10.7554/eLife.43803.019

The following source data and figure supplements are available for figure 4:

**Source data 1.** Application of cNMF to mouse visual cortex data.
DOI: https://doi.org/10.7554/eLife.43803.023
**Figure supplement 1.** Diagnostic plot for cNMF on the *Hrvatin et al. (2018)* visual cortex dataset.
DOI: https://doi.org/10.7554/eLife.43803.020
**Figure supplement 2.** Comparison of cNMF usages with the visual cortex cell-type clusters from *Hrvatin et al. (2018)*.
DOI: https://doi.org/10.7554/eLife.43803.021
*Figure 4 continued on next page*

*Figure 4 continued*

**Figure supplement 3.** Comparison of GEPs identified in the *Hrvatin et al. (2018)* and *Tasic et al. (2016)* visual cortex datasets.

DOI: https://doi.org/10.7554/eLife.43803.022

*Supplementary file 3*) which further suggests its interpretation as a program involved in the formation or regulation of synapses. The last activity program (labeled Other) was characterized by high expression of the maternally expressed long non-coding RNA *Meg3* (*Yan et al., 2016*) and other genes that are associated with cerebral ischemic injury (e.g. *Glg1* [*Zhang et al., 1996*], *Rtn1* [*Gong et al., 2017*]). Our functional interpretations of the novel activity programs are speculative, but they highlight the ability of cNMF to identify intriguing GEPs in an unbiased fashion.

## Discussion

In this study, we distinguished between cell type (identity) and cell type independent (activity) gene expression programs (GEPs) to motivate our use of matrix factorization, which represents cells as linear combinations of multiple GEPs. However, we note that some biological programs are not neatly classified as either identity or activity GEPs. For example, cell states reflecting oncogenic transformation, or a cell's position along a morphological gradient blur the distinction between identity and activity. In addition, stochastic fluctuations in individual transcription factors could result in coordinated gene expression changes (*Thattai and van Oudenaarden, 2001*) and might be better described as a third program category, rather than as an identity or activity GEP. While the identity/activity distinction might not be appropriate in every case, matrix factorization should, in principle, be appropriate for representing all gene expression states that can be reasonably approximated as a linear mixture of programs.

Furthermore, in this study, we have provided an empirical foundation for the use of matrix factorization to simultaneously infer identity and activity programs from scRNA-Seq data. We first show with simulations that despite their simplifying assumptions, ICA, LDA, and NMF (but not PCA) can infer components that align well with GEPs. However, due to the stochastic nature of these algorithms, the interpretability and accuracy of individual solutions can be low. This led us to develop a consensus approach that empirically increased the accuracy and robustness of the solutions. cNMF inferred the most accurate identity and activity programs of all the methods we tested. Moreover, it yielded results in interpretable units of gene expression (transcripts per million) and could accurately infer the percentage of each cell's expression that was derived from each GEP. These properties made it the most promising approach for GEP inference on real datasets.

We then explored the utility of cNMF on real data, recapitulating known GEPs, identifying novel ones, and further characterizing their usage. We first validated cNMF with several expected activity programs serving as positive controls. We then identified several unexpected but highly plausible programs, a hypoxia program in brain organoids and a depolarization-induced activity program in untreated neurons. Finally, we identified three novel programs in visual cortex neurons that we speculate may correspond to a neurosecretory phenotype, new synapse formation, and a stress response program. Beyond simply discovering activity programs, cNMF clarified the underlying cell types in these datasets by disentangling activity and identity programs from the mixed single-cell profiles. For example, we found that a brain organoid subpopulation that was initially annotated as proliferative precursors actually includes replicating cells of several cell types such as an immature skeletal muscle cell that is differentiating into slow-twitch and fast-twitch muscle populations. Furthermore, joint analysis of identity and activity GEPs allowed us to quantify the relative prevalence of activities across cell types. For example, we found in the visual cortex data that one depolarization-induced late response program was predominantly expressed in neurons of superficial cortical layers, while the other was mainly expressed in deeper layers. This suggests that an anatomical or developmental factor may underlie variability in the response. While commonly used approaches based on clustering or pseudotemporal ordering of cells are poorly suited to achieve such insights from single-cell data, these findings emerge naturally from our matrix factorization approach.

We have made our tools and analyses easily accessible so that researchers can readily use cNMF and further develop on the approach. We have deposited all the cNMF code on Github https://github.com/dylkot/cNMF/ (*Kotliar, 2019*; copy archived at https://github.com/elifesciences-

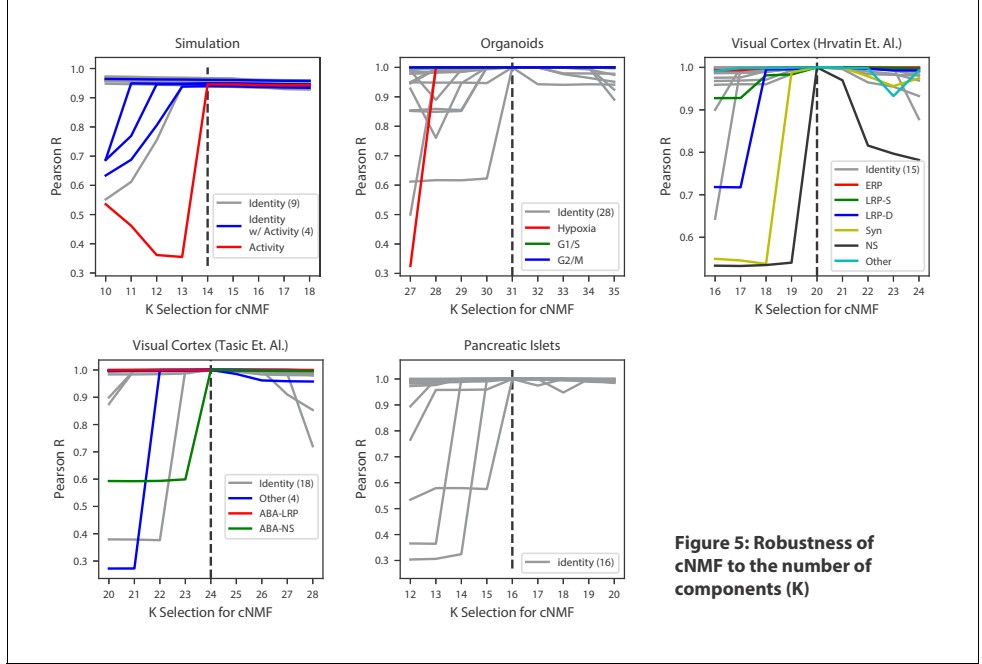

**Figure 5: Robustness of cNMF to the number of components (K)**

**Figure 5.** Robustness of cNMF to the number of components (K). Line plots of the maximum Pearson correlation between each of the cNMF components presented in the main analysis, and the cNMF components that result from multiple choices of K. For the simulated data, for which we have access to ground truth, we plot the correlation between the inferred components for each choice of K and the ground truth 14 components. We highlight components corresponding to activity GEPs with distinct colors and denote the number of identity GEPs contained on the same plot in parenthesis in the legend. A dashed line indicates the K choice that was presented in the main analysis. Pearson correlations are computed considering only the 2000 most over-dispersed genes and on vectors normalized by the computed sample standard deviation of each gene.

DOI: https://doi.org/10.7554/eLife.43803.024

The following source data and figure supplements are available for figure 5:

**Source data 1.** Application of cNMF to pancreas data and analysis of robusness to the choice of K.
DOI: https://doi.org/10.7554/eLife.43803.027

**Figure supplement 1.** Characterization of GEP usage across biological replicates.
DOI: https://doi.org/10.7554/eLife.43803.025

**Figure supplement 2.** Comparison of cNMF usages with the published cell-type clusters from *Baron et al. (2016)*.
DOI: https://doi.org/10.7554/eLife.43803.026

publications/cNMF) and have made available all of the analysis scripts for figures contained in this manuscript on Code Ocean (https://doi.org/10.24433/CO.9044782e-cb96-4733-8a4f-bf42c21399e6) for easy exploration and re-execution.

As others apply this approach, one key consideration will be the choice of the three input parameters required by cNMF: the number of components to be found (K), the percentage of replicates to use as nearest neighbors for outlier-detection, and a distance threshold for defining outliers. While the choice of K must ultimately reflect the resolution desired by the analyst, we propose two simple decision aids based on (1) considering the trade-off between factorization stability and reconstruction error and (2) looking at the proportion of variance explained by K principal components to estimate the dimensionality of the data (*Figure 2—figure supplement 3*, *Figure 3—figure supplement 1*, *Figure 4—figure supplement 1*). In addition, we noticed that choosing consecutive values of K primarily influenced individual components at the margin, suggesting that cNMF may be robust to this choice within a reasonable range of options (*Figure 5* and 'Choosing the number of components' section of the Materials and methods).

The additional two parameters allow users to optionally identify outlier replicates to filter before averaging across replicates. This improves overall accuracy by removing infrequent solutions that often represent merges or splits of the true GEPs. Using 30% of the number of replicates as nearest neighbors worked well for all datasets we analyzed, and an appropriate outlier distance threshold

was clear in our applications based on the long tail in the distance distribution (*Figure 2—figure supplement 3*, *Figure 3—figure supplement 1*, *Figure 4—figure supplement 1*).

Our approach is an initial step toward disentangling identity and activity GEPs, and will benefit from subsequent development. For example, cNMF does not specifically address the count nature of gene expression. Recently developed statistical frameworks that address these aspects of scRNA-Seq data such as Hierarchical Poisson Factorization (*Levitin et al., 2018*) may therefore increase the accuracy of GEP inference. In addition, NMF often yields low but non-zero usages for many GEPs even though we expect most cells to express a small number of identity and activity GEPs. This lack of sparsity is likely due to over-fitting and could be addressed by adding regularization to the model (*Taslaman and Nilsson, 2012*). Such refinements and any new matrix factorization that relies on stochastic optimization can be readily combined with our consensus approach to potentially improve accuracy and interpretability.

A more fundamental limitation of matrix factorizations, including cNMF, is the built-in assumption that cells can be modeled as linear combinations of GEPs. Notably, this precludes modeling of transcriptional repression, where one or more genes that would be induced by one GEP are significantly reduced in expression when a second repressing GEP is active in the same cell. To our knowledge, such relationships have not been represented in a matrix factorization framework, but they may be easier to incorporate in new classes of latent variable models such as variational autoencoders (VAEs) (*Ding et al., 2018*; *Grønbech et al., 2018*). VAEs represent cells in a highly flexible latent space that can capture non-linearities and interactions between latent variables. However, while the latent variables are designed to facilitate accurate reconstruction of the input gene expression data, it remains to be shown whether they can be directly or indirectly interpreted as distinct GEPs and GEP usages. For the foreseeable future, there may be a trade-off between the flexibility of these models and the difficulty in training them and interpreting their output.

With ongoing technological progress in RNA capture efficiency and throughput, scRNA-Seq data is likely to become richer and more expansive. This will make it possible to detect increasingly subtle GEPs, reflecting biological variability in cell types, cell states, and activities. Here, we have demonstrated a computational framework that can be used to infer such GEPs directly from the scRNA-Seq data without the need for experimental manipulations, providing key insights into the behavior of cells and tissues.

## Materials and methods

### Simulations

Our simulation framework is based on Splatter (*Zappia et al., 2017*) but is re-implemented in Python and adapted to allow simulation of doublets and activity programs. 14 gene-expression programs $Z \in N^M$ (13 identity programs $Z_1 \ldots Z_{13}$ and 1 activity program $Z_a$, each a vector of M genes) were simulated as in Splatter. Cells were then randomly assigned to an identity program with equal probability for each class. 30% of cells of four cell types were randomly selected to express the activity program at a usage $\phi_i$, uniformly distributed between 10% and 70%. In the Splatter notation, the pre-trended mean gene-expression profile $\lambda'_i$ for each cell i=1... 15,000 was computed as the weighted sum of the identity and the activity program:

$$\lambda'_i = L_i \big( \phi_i Z_a + (1 - \phi_i) Z_{I(i)} \big)$$

where $L_i$ is the simulated library size for a cell, $I(i)$ is the identity program assignment for cell i, and $\phi_i = 0$ for cells that do not express the activity GEP and $\phi_i \sim Uniform(0.1, 0.7)$ for cells that do.

Doublets were constructed by randomly sampling pairs of cells, summing their simulated count vectors, and randomly down-sampling the counts until the total number of counts equaled the maximum of the library sizes of the original two cells. We simulated 25,000 genes, 1000 of which were associated with the activity program. The probability of a gene being differentially expressed in a cell identity program was set to 2.5%. The differential expression scale parameter was 1.0 for all simulations and the location parameter was either 1.0, 0.75, or 0.5 to simulate different signal to noise levels. Other splatter parameters were: lib.loc = 7.64, libscale = 0.78, mean_rate = 7.68, mean_shape = 0.34, expoutprob = 0.00286, expoutloc = 6.15, expoutscale = 0.49, diffexpdownprob = 0,

bcv_dispersion = 0.448, bcv_dof = 22.087. These values were inferred from 8000 randomly sampled cells of the *Quadrato et al. (2017)* organoid dataset using Splatter. A differential expression location parameter of 1.0 were used for the 50% doublet simulation and all other parameters were kept the same (*Figure 2—figure supplement 7*). Differential expression location parameters of 1.0 and 2.0 were used for the two variable cell-type proportion simulations, and cell-type assignments were sampled from a multinomial with parameters (0.215, 0.210, 0.195, 0.130, 0.0924, 0.0328, 0.028, 0.0269, 0.0149, 0.0119, 0.0114, 0.009, 0.009, 0.008, 0.006) based on the proportions of neuronal cell-type clusters in *Hrvatin et al. (2018)* (*Figure 2—figure supplement 8*). All other parameters were as described above.

## Data preprocessing

For each dataset, we removed cells with fewer than 1000 unique molecular identifiers (UMIs) detected. We also filtered out genes that were not detected in at least 1 out of 500 cells. We denote this filtered count matrix as $C_{ij}$ (i=1 … N cells, j=1 … M genes). We denote matrices in bold.

We then selected the H most over-dispersed genes as determined by the v-score (*Klein et al., 2015*) for input to cNMF. H was set to 2000 for all datasets analyzed in this manuscript. It is essential to select a subset of over-dispersed genes prior to normalization because afterwards, variation in lower-variance genes due to noise will be on the same scale as biologically meaningful variation. 2000 reflects a trade-off between including enough genes to detect subtle biological signals, and speeding up computation by not including too many extraneous genes.

Each of the H over-dispersed genes was then scaled to unit variance before running cNMF resulting in a normalized matrix:

$$\tilde{C} = \left[ \cdots \quad \frac{C_j}{\sigma(C_j)} \quad \cdots \right] \text{ for genes } j \in \{H \text{ overdispersed genes}\}$$

where $C_j$ denotes the jth column of **C**, $\sigma(C_j)$ denotes the sample standard deviation of $C_j$, and $\{H \text{ overdispersed genes}\}$ denotes the previously selected set of over-dispersed genes. This variance-scaling is similar to the log transformation that is commonly applied to scRNA-Seq data in that it ensures genes on different expression scales contribute comparable amounts of information to the GEP inference. However, we prefer variance scaling because log transformation requires addition of an arbitrary pseudo-count which can substantially impact downstream analysis (*William Townes et al., 2019*). In addition log-transformation renders fold-changes of 0.1 to 1, and 100 to 1000 equivalent in absolute terms. This is undesirable given the current resolution of scRNA-Seq data because the former change can frequently be attributable to noise while we have much greater confidence in the biological significance of the latter. Variance scaling the data avoids the need for any modulation to the shape of a gene's distribution and avoids the need for addition of an arbitrary pseudocount. Finally, addition of components in log expression space corresponds to multiplication in raw expression space. We believe additivity to be a more appropriate model than multiplicativity for most GEPs, and therefore do not log-transform the data prior to running cNMF. We do not mean center the genes so as to preserve the non-negativity of the expression data which is a requirement for NMF.

Note that we do not perform any cell count normalization (I.e. normalization of the rows of **C** prior to cNMF). This is because cells with more counts can contribute more information to the model. Technical variation in transcript abundances across cells are captured in the usage matrix rather than the component matrix. However, for the *Tasic et al. (2016)* dataset, which is based on full-transcript sequencing rather than digital UMI counting, we variance-normalized high-variance genes from the TPM matrix directly rather than from the raw count matrix as in the other datasets.

As a final step in cNMF, the consensus programs can be re-fit in physically meaningful (non-normalized) biological units of the user's choice, such as transcripts per million and including all genes (not just the H over-dispersed ones). See below for details.

## Consensus non-negative matrix factorization (cNMF)

We use non-negative matrix factorization implemented in scikit-learn version 20.0 (RRID:SCR_002577) with the default parameters except for random initialization, tolerance for the stopping condition of $10^{-4}$, and a maximum number of iterations of 400.

R replicates of NMF are run on the same normalized dataset with the same number of components K but with different randomly selected seeds, resulting in R instances of usage matrices $U^{(r)}$ (N cells x K programs) and program matrices $G^{(r)}$ (K programs x H genes):

$$U^{(r)} x\, G^{(r)} \approx \tilde{C} \quad for\ r = 1 \dots R$$

For each replicate r, the rows of $G_r$ are normalized to have l2 norm of 1:

$$\tilde{G}^{(r)} = \begin{bmatrix} \frac{G_1^{(r)}}{||G_1^{(r)}||_2} \\ \vdots \\ \frac{G_k^{(r)}}{||G_k^{(r)}||_2} \\ \vdots \\ \frac{G_K^{(r)}}{||G_K^{(r)}||_2} \end{bmatrix}$$

where $G_k^{(r)}$ is the kth row of the programs matrix for the rth NMF replicate $G^{(r)}$ and $||\ ||_2$ denotes the l2 norm.

The component matrices from each replicate are then concatenated vertically into a single RK x H dimensional matrix, **G**, where each row is a component from one replicate:

$$\mathbf{G} = \begin{bmatrix} \tilde{G}^{(1)} \\ \vdots \\ \tilde{G}^{(r)} \\ \vdots \\ \tilde{G}^{(R)} \end{bmatrix}$$

Components with high mean Euclidean distance from their L nearest neighbors are then filtered out as below:

$$\mathrm{L} = \rho \mathrm{R}$$

$$D(G_l,\ \mathrm{L}) = \frac{1}{L} \sum_{G_n \in N_L(G_l)} ||G_l - G_n||_2$$

$$G^{(f)} = \begin{bmatrix} \dots \\ G_l \\ \dots \end{bmatrix} for\ l = 1 \dots Rk\ if\ D(G_l,\ \mathrm{L}) < \tau$$

where $G_l$ is the lth row of **G**, $N_L(G_l)$ is the set of L nearest neighbors of $G_l$ and $G^{(f)}$ is the matrix of rows that passed the L nearest neighbors distance threshold filter.

Two user-specified parameters, ρ and τ, determine which replicate components are filtered out and which are kept. ρ denotes the fraction of NMF replicates to be used as nearest neighbors. Intuitively, ρ can be thought of as the fraction of replicates that must yield a component approximately matching a program in order for that program to be kept by cNMF. τ is a distance threshold that determines how close a component must be to its nearest neighbors in Euclidean space to be considered 'approximately matching'.

Our choice of ρ and τ were guided by inspection of the clustergram and histogram output by cNMF with the goal of filtering out outlier components and yielding clean correlation blocks on the clustergrams (*Figure 2—figure supplement 3*, *Figure 3—figure supplement 1*, *Figure 4—figure supplement 1*). We set ρ=0.3 for all datasets analyzed in this manuscript which

reflects a tolerance to identify components that occur approximately in 30% or more replicates. We find this to be an appropriate default setting. We set $\tau$ = 0.03, 0.10, 0.08, 0.10, and 0.04 for the simulation, organoid, main visual cortex (*Hrvatin et al., 2018*), secondary visual cortex (*Tasic et al., 2016*), and the Pancreatic islets datasets (discussed in the supplementary note on between-sample variability) based on truncating the long tail of the distance to KNN histogram.

Next, the rows of $\boldsymbol{G}^{(f)}$ are clustered using KMeans with the Euclidean distance metric and the same number of clusters (K) as the number of components for the NMF runs. This defines sets $A_k = \{rows\ l\ assigned\ to\ cluster\ k\}$ containing the indices of the rows of $\boldsymbol{G}^{(f)}$ that are assigned to the kth cluster.

Each cluster of replicate components is then collapsed down to a single consensus vector by taking the median value for each gene across components in a cluster:

$$\boldsymbol{G}_{kj}^{(c)} = median\left(\left\{ G_{lj}^{(f)}\ for\ l \in A_k \right\}\right)$$

for *l* indexing over rows of $\boldsymbol{G}^{(f)}$ and j indexing over columns (genes), and with the median taken separately for each gene j. This defines a KxH consensus programs matrix $\boldsymbol{G}^{(c)}$ where the (c) superscript denotes consensus. The merged GEP components are then l1 normalized:

$$\tilde{\boldsymbol{G}}^{()} = \begin{bmatrix} \frac{G_1^{(c)}}{||G_1^{(c)}||_1} \\ \vdots \\ \frac{G_k^{(c)}}{||G_k^{(c)}||_1} \\ \vdots \\ \frac{G_K^{(c)}}{||G_K^{(c)}||_1} \end{bmatrix}$$

where $G_k^{(c)}$ is the kth row of $\boldsymbol{G}^{(c)}$ and $||\ ||_1$ denotes the l1 norm. A consensus usage matrix is then fit by running one last iteration of NMF with the component matrix fixed to $\tilde{\boldsymbol{G}}^{(c)}$. This amounts to fitting non-negative least squares regressions of each cell's normalized expression profile $\tilde{C}_i$ against $\tilde{\boldsymbol{G}}^{(c)}$ by solving the following optimization:

$$\min_{U_{i1}\ldots U_{iK} \geq 0} ||\ \tilde{C}_i - \sum_{k=1}^{K} U_{ik}\ \tilde{G}_k^{(c)}\ ||_2$$

where $\tilde{G}_k^{(c)}$ is the H-dimensional normalized consensus program vector for the kth GEP, i indexes over cells, and we are maximizing with respect to GEP usage values $U_{i1}\ldots U_{iK}$ which are constrained to be non-negative. We concatenate all these coefficients into a consensus usage matrix $\boldsymbol{U}^{(c)}$:

$$\boldsymbol{U}^{(c)} = \begin{bmatrix} U_{11} & \cdots & U_{1K} \\ \vdots & \ddots & \vdots \\ U_{N1} & \cdots & U_{NK} \end{bmatrix}$$

and normalize it so that the usage values for each cell sum to 1:

$$\tilde{\boldsymbol{U}}^{()} = \begin{bmatrix} \frac{U_1^{(c)}}{||U_1^{(c)}||_1} \\ \vdots \\ \frac{U_i^{(c)}}{||U_i^{(c)}||_1} \\ \vdots \\ \frac{U_N^{(c)}}{||U_N^{(c)}||_1} \end{bmatrix}$$

where $U_i^{(c)}$ is the ith row of the consensus usage matrix. With this normalized, consensus usage

matrix fixed, final program estimates can be computed in desired units, and for all genes—including genes that were not initially included among the over-dispersed set. This is done by running a last iteration of NMF with the usage matrix fixed as $\tilde{U}^{(c)}$ and the input data reflecting the desired final units. To convert the estimated programs to TPM units and to obtain program vectors spanning the full set of input genes, we refit against the matrix of TPM values, **T**:

$$
\boldsymbol{T} = \begin{bmatrix} \frac{10^6 C_1}{||C_1||_1} \\ \vdots \\ \frac{10^6 C_i}{||C_i||_1} \\ \vdots \\ \frac{10^6 C_N}{||C_N||_1} \end{bmatrix}
$$

$$
\min_{G_{j1}^{(TPM)}...G_{jK}^{(TPM)} \geq 0} || T_j - \sum_{k=1}^{K} \tilde{U}_{:,k}^{(c)} \, G_{jk}^{(TPM)} ||_2
$$

where $T_j$ is the N-dimensional TPM profile for the jth gene, $\tilde{U}_{:,k}^{(c)}$ is the N-dimensional normalized consensus profile for the kth GEP over all cells, and $G_{jk}^{(TPM)}$ is an estimated coefficient reflecting how much the TPM of gene j is expected to increase per a unit increase in usage of GEP k, if all other usages were held constant. Note that the TPM matrix **T** is calculated using a raw count matrix **C** that includes all genes, even those that were filtered out for falling below the count threshold. We repeat this for all M genes in the filtered count matrix and combine allthese coefficients into a consensus program matrix:

$$
\boldsymbol{G^{(TPM)}} = \begin{bmatrix} G_{11}^{(TPM)} & \cdots & G_{1M}^{(TPM)} \\ \vdots & \ddots & \vdots \\ G_{K1}^{(TPM)} & \cdots & G_{KM}^{(TPM)} \end{bmatrix}
$$

## Identification of marker genes

We identify marker genes (genes that are statistically associated with each GEP) using multiple least squares regression of normalized (z-scored) gene expression against the consensus GEP usage matrix. This amounts to finding the genes that have higher than average expression for cells that use a specific GEP. We compute the z-score of the TPM profile like so:

$$
Z_j = \frac{T_j - \mu_j}{\sigma_j}
$$

where $T_j$ is the TPM profile of the jth gene, $\mu_j$ is the sample mean, and $\sigma_j$ is the sample standard deviation of $T_j$. Then we fit $\beta_{1j}... \beta_{kj} ...\beta_{Kj}$, coefficients reflecting the association between GEP k and gene j using ordinary least squares regression by solving the minimization:

$$
\min_{\beta_{1j}... ...\beta_{Kj}} || Z_j - \sum_{k=1}^{K} U_k^{(c)} \, \beta_{1k} ||^2
$$

where $U_k^{(c)}$ is the kth column of the un-normalized consensus usage matrix. The regression coefficients $\beta_{kj}$ can then be interpreted as by how many standard deviations the expression of gene j should increase for an additional count of usage being attributed to GEP k. We regress against z-scored expression values rather than the un-normalized expression values so that the coefficients will be comparable between genes expressed on different scales. For discrete clustering methods, the usage matrix is a binary indicator matrix containing a 1 for the cluster (column) each cell is assigned to, and a 0 for all other columns. In the discrete clustering context, $\beta_{kj}$ can be interpreted as the average expression of gene j in cells assigned to cluster k. Identifying marker genes through multivariate regression in this fashion, rather than through separate tests for each GEP, reduces the risk of confounding that can occur when GEPs tend to be expressed in the same cells. For example,

if an activity GEP is predominantly expressed in cells of a specific cell-type, it avoids misattributing activity genes to the identity program of that cell-type and vice versa.

We note that because gene-expression data is not normally distributed, the residuals of the regression will not be normal, which violates an assumption of OLS regression. However, the coefficient estimates will still be unbiased even if normality is violated. In practice, we do not use the p-values of the regressions at an any point in our analysis as those can be inaccurate due to non-normality. We recommend testing for gene-set enrichment on regression coefficients directly (as we discuss below) rather than setting thresholds on regression P-values.

## Choosing the number of components

Determining the number of components (K) to use for cNMF is an important but challenging step without a simple approach that can work for all datasets and applications. We use two diagnostic plots to help guide this decision. The first plot shows the stability of the solution (as captured by the silhouette score) and the Frobenius reconstruction error as a function of K as described previously in *Alexandrov et al. (2013)*. However, unlike in *Alexandrov et al. (2013)*, we run NMF on normalized data matrices rather than count matrices and therefore do not resample counts but simply repeat NMF with different randomly selected seeds. We compute the Frobenius error using the consensus NMF solution but without any outlier filtering. We also use the Euclidean distance on l2 normalized components as the metric for the silhouette score rather than Cosine distance. Silhouette score is calculated using the Scikit-learn version 20.0 silhouette_score function. We parallelized the individual factorization steps over cores on a multi-core virtual machine using GNU Parallel (*Tange, 2011*).

As another approach to confirm the appropriateness of our choice of K, we use scree plots which depict the proportion of variance explained per principal component (*Figure 2—figure supplement 3*, *Figure 3—figure supplement 1*, *Figure 4—figure supplement 1*). This is motivated by the fact that choosing the optimal number of principal components and choosing the number of NMF components can both be framed as estimating the rank for a low-dimensional representation of the input data. As a consequence of the Eckart Young Mirsky theorem, PCA necessarily provides the matrix factorization with minimum Frobenius reconstruction error for any choice of K (*Eckart and Young, 1936*) and we also use the Frobenius error in our NMF model. Because principal components are orthogonal to each other, and loadings of NMF components can never be negative, K principal components will always span a larger sub-space than K NMF components. This suggests that the optimal number of NMF components will likely not be smaller than the optimal number of PCs. The scree plot is a commonly used tool to estimate the number of principal components and we use it to help guide the number of NMF components as well.

We note that these two plots merely provide a general aid for the choice of K and we considered the biological interpretability of factors found from several choices of K before proceeding. We do not recommend necessarily using the maximum stability solution of the error vs. stability plot as this can frequently miss true biological signal and, indeed would have led to the incorrect choice for the simulated data (*Figure 2—figure supplement 3*).

Given the uncertainty of the choice of K, we confirmed that the conclusions of this manuscript are robust to this decision. When we varied K within a range of ±four around the choice used in the manuscript, we found approximately the same core set of GEPs with a single new GEP being discerned with each consecutive step in K. For each step below the selected K, approximately a single GEP was lost, but for choices above the selected K, components approximately matching the original K programs (I.e. with Pearson correlation >0.7) were found (*Figure 5*). This suggests that cNMF yields relatively stable solutions for a moderate range of K values.

## Comparison of cNMF with other methods

We compared cNMF with consensus and standard versions of LDA and ICA as well as with PCA, Louvain clustering and a hard clustering based on assignment of cells to their ground-truth labels. We used the implementations of LDA, ICA, and PCA in scikit-learn and the implementation of Louvain clustering in scanpy (*Wolf et al., 2018*). For ICA, we used the FastICA implementation with default options for all the parameters. For LDA, we used the batch algorithm and all other parameters as defaults. We defined the consensus estimates across 200 replicates in the same way as for cNMF but

with a slight modification for ICA. Because ICA is under-determined with respect to the signs of the solutions, some iterations will yield a given component pointed in one direction while others produce approximately the same component but pointed in the opposite direction (multiplied by $-1$). Therefore, we aligned the orientation of components from across replicates by identifying any components whose median usage across all cells was positive and scaled those and the corresponding usages by $-1$.

For Louvain clustering, we used 14 principal components to compute distances between cells and used 200 nearest neighbors to define the KNN graph. We chose 14 principal components based on the fact that the data was simulated based on a 14-dimensional basis and, therefore, the biological variation in the data can be captured by 14 PCs and subsequent components correspond to noise. This choice is also justified by choosing the elbow on scree plot in *Figure 2—figure supplement 3*. We used 200 nearest neighbors for the clustering as this is a relatively large number to minimize variance but it is still smaller than the smallest discrete population (0.3*15,000*(1/13)=346 cells from a specific cell-type that expresses the activity program).

For ground-truth assignment clustering, we assigned each cell to a cluster defined by its true identity program, except for cells which had greater that 40% usage of the activity program, which we assigned to an activity program cluster. Then we determined a GEP corresponding to each cluster as the mean TPM value for each gene over cells in the cluster.

To evaluate the accuracy of these various methods, we first calculated the z-score coefficient for associating each gene with each program as described above. We then calculated sensitivity and false discovery rate (FDR) for each threshold on those coefficients and plotted those as an ROC-curve, except with FDR on the X-axis instead of false positive rate. For this evaluation, we considered a gene as truly associated with a GEP if it had a ground-truth fold-change of >= 2 and truly unassociated with a GEP if the ground-truth fold-change was 1. Genes with a fold-change between 1 and 2 were ignored for this evaluation.

## Testing enrichment of genesets in programs

We used the z-score regression coefficients identified as above as input for a one-sided Mann Whitney U Test (with tie correction) comparing the median of genes in each geneset to those of genes not in the geneset. We first floored all negative coefficients to equal zero prior to the test. Coefficients less than 0 indicate genes that are expressed at higher levels in cells that do not use the GEP (all other things equal) than in cells that do. We floor these values so that variation in genes that are not directly part of a GEP (which can make up the majority of genes) do not substantially impact the Mann-Whitney statistic for that GEP.

## Data availability

The organoid data described in the manuscript is accessible at NCBI GEO accession number GSE86153. However, we obtained the clustering and unnormalized data by request from the authors. The visual cortex datasets used for *Figure 3* are accessible at NCBI GEO, accession numbers GSE102827 and GSE71585.

## Code availability

Code for running cNMF is available on Github https://github.com/dylkot/cNMF, as is code for simulating data with doublets and activity programs https://github.com/dylkot/scsim (*Kotliar and Eraslan, 2019*; copy archived at https://github.com/elifesciences-publications/scsim).

All datasets and analysis used in this manuscript are available for download, exploration and re-execution on Code Ocean: https://doi.org/10.24433/CO.9044782e-cb96-4733-8a4f-bf42c21399e6.

## Supplementary note - Analysis of between-sample variability

We sought to understand how variability between sample replicates and batches would impact the results of cNMF. We therefore considered how GEP usage varies across replicates in the primary datasets analyzed in this manuscript, as well as in a previously published scRNA-Seq dataset of human pancreatic islets with noted batch-effect (*Baron et al., 2016*).

First, we analyzed the aggregate GEP usage of cells in organoid replicates in the *Quadrato et al. (2017)* data, and mouse replicates in the *Hrvatin et al. (2018)* visual cortex data. For this purpose,

we defined the aggregate GEP profile of a replicate as the sum of the GEP usage of all cells derived from that replicate. The visual cortex data showed relative uniformity of GEP usage across mouse replicates, with the only clear pattern being the expected association between depolarization-induced GEPs and mice treated with the stimulus (*Figure 5—figure supplement 1a* - left). By contrast, there was significant variability between organoids in the *Quadrato et al. (2017)* data that was primarily associated with the bioreactors in which the organoids were grown (*Figure 5—figure supplement 1b* - left). This variability was discussed in the original manuscript and validated using immunohistochemistry, and thus represents true biological signal that we would hope for cNMF to discern.

We also considered whether any GEPs could be attributed to just one or a small number of replicates which could suggest that they are not reproducible within the experiment. We therefore looked at what percentage of the aggregate usage of a GEP derived from cells in each replicate. We found that each GEP contributed to cells from multiple independent replicates in both datasets (*Figure 5—figure supplement 1*, right panels). No GEP derived more than 15% of its usage from a single replicate in the visual cortex data or more than 45% of its usage from a single replicate in the organoid data. Furthermore, each organoid GEP was the maximum contributing GEP for a cell in at least six distinct organoid replicates, and each visual cortex GEP was the maximum contributor for a cell in at least 10 distinct mouse replicates. This supports our conclusion that the inferred GEPs represent reproducible signals within the primary organoid and visual cortex datasets.

We also analyzed a human pancreatic islet scRNA-Seq dataset where variability between four donors resulted in more substantial batch-effects to see how that would impact the behavior of cNMF (*Baron et al., 2016*). Applied to this dataset of 10,939 cells, cNMF identified 16 GEPs that corresponded well with the cell-type clusters described in the initial publication (*Figure 5—figure supplement 2*). Our application of cNMF failed to identify GEPs corresponding to a few cell-types described in *Baron et al. (2016)* (e.g. cells distinguished as delta and gamma cell-types were assigned the same GEP). However, many of the cell types that were missed by cNMF were only distinguished through iterative sub-clustering in the initial publication, which we did not attempt.

Notably, we identified multiple GEPs for many cell-type clusters that corresponded to 'donor of origin.' For example, we identified separate GEPs corresponding to acinar cells derived from donors 1 and 3, and acinar cells derived from donors 2 and 4, and similarly for alpha, ductal, and stellate cells. One potential contributor to the batch-effect could be that donors 1 and 3 were male and donors 2 and 4 were female. Consistent with this, we noticed that among the genes that were most differentially expressed between donors 1 and 3 compared to donors 2 and 4 in alpha, beta, and acinar cells were XIST on the X chromosome and RPSY1 on the Y chromosome (linear regression F-test p-values$<5\times10^{-243}$ for for XIST and p-values$<4\times10^{-145}$ for RPSY1 for all 3 cell-types tested). But in general, the fact that cNMF is discerning multiple GEPs for the same cell-types suggests that technical sources of variation such as batch-effect can confound the identification of identity and activity GEPs.

In this instance, cNMF did not learn a single GEP for each donor (I.e. batch) but rather identified multiple hybrid identity-donor GEPs corresponding to individual cell-types derived from distinct sets of donors. This is likely due to the fact that the batch effect modulated the expression of different sets of genes in different cell-types, and therefore, no single shared 'batch-effect' GEP could capture the impact on each cell-type. To avoid incorporating variation between batches into the inferred GEPs for datasets containing significant batch-effect, batch-effect correction can be performed prior to running cNMF.

## Acknowledgements

We thank Allon Klein, Samuel Wolock, Aubrey Faust, Chris Edwards, Stephen Schaffner, Eric Lander, the CGTA discussion group, and members of the Sabeti Laboratory for useful discussions and feedback on the manuscript. We thank the Arlotta, Greenberg, and Zeng laboratories for generating the primary datasets we analyze in this manuscript. The project described was supported by award Number R01AI099210 from the National Institute of Allergy and Infectious Disease, HHSF223201810172C from the U.S. Food and Drug Administration, and T32GM007753 from the National Institute of General Medical Sciences. The content is solely the responsibility of the authors

and does not necessarily represent the official views of the National Institute of General Medical Sciences, Food and Drug Administration, or the National Institutes of Health.

## Additional information

### Funding

| Funder | Grant reference number | Author |
|---|---|---|
| National Institute of General Medical Sciences | T32GM007753 | Dylan Kotliar<br>Adrian Veres<br>M Aurel Nagy<br>Eran Hodis |
| National Institute of Allergy and Infectious Diseases | R01AI099210 | Pardis C Sabeti |
| U.S. Food and Drug Administration | HHSF223201810172C | Dylan Kotliar<br>Pardis C Sabeti |

The funders had no role in study design, data collection and interpretation, or the decision to submit the work for publication.

### Author contributions

Dylan Kotliar, Conceptualization, Resources, Data curation, Software, Formal analysis, Investigation, Methodology; Adrian Veres, Conceptualization, Software, Formal analysis, Investigation, Methodology; M Aurel Nagy, Resources, Formal analysis; Shervin Tabrizi, Eran Hodis, Investigation, Helped analyze data during an early version of the project that shaped the specifics of the methodology and analysis; Douglas A Melton, Pardis C Sabeti, Supervision, Funding acquisition, Writing—original draft

### Author ORCIDs

Dylan Kotliar (ID) https://orcid.org/0000-0002-7968-645X
M Aurel Nagy (ID) https://orcid.org/0000-0003-4608-1152
Shervin Tabrizi (ID) https://orcid.org/0000-0003-2780-8432
Douglas A Melton (ID) http://orcid.org/0000-0002-1623-5504

### Decision letter and Author response

Decision letter https://doi.org/10.7554/eLife.43803.043
Author response https://doi.org/10.7554/eLife.43803.044

## Additional files

### Supplementary files

• Supplementary file 1. Brain organoid GEP genescores.
DOI: https://doi.org/10.7554/eLife.43803.028

• Supplementary file 2. Visual cortex GEP genescores.
DOI: https://doi.org/10.7554/eLife.43803.029

• Supplementary file 3. Novel activity GEP enrichments.
DOI: https://doi.org/10.7554/eLife.43803.030

• Transparent reporting form
DOI: https://doi.org/10.7554/eLife.43803.031

### Data availability

All of the analyzed real datasets are publicly available and the relevant GEO accession codes are included in the manuscript. All of the simulated and real data can be accessed through Code Ocean at the following URL: https://doi.org/10.24433/CO.9044782e-cb96-4733-8a4f-bf42c21399e6. cNMF code is available on Github https://github.com/dylkot/cNMF/ (copy archived at https://github.com/elifesciences-publications/cNMF).

The following dataset was generated:

| Author(s) | Year | Dataset title | Dataset URL | Database and Identifier |
|---|---|---|---|---|
| Kotliar D, Veres A, Nagy MA, Tabrizi S, Hodis E, Melton DA, Sabeti PC | 2019 | Identifying Gene Expression Programs of Cell-type Identity and Cellular Activity with Single-Cell RNA-Seq | https://doi.org/10.24433/CO.9044782e-cb96-4733-8a4f-bf42c21399e6 | Code Ocean, 10.24433/CO.9044782e-cb96-4733-8a4f-bf42c21399e6 |

The following previously published datasets were used:

| Author(s) | Year | Dataset title | Dataset URL | Database and Identifier |
|---|---|---|---|---|
| Quadrato G, Nguyen T, Macosko EZ, Sherwood JL, Berger D, Maria N, Scholvin J, Goldman M, Kinney J, Boyden E, Lichtman J, Williams ZM, McCarroll SA, Arlotta P | 2017 | Cell diversity and network dynamics in photosensitive human brain organoids. | https://www.ncbi.nlm.nih.gov/geo/query/acc.cgi?acc=GSE86153 | Gene Expression Omnibus, GSE86153 |
| Hrvatin S, Hochbaum DR, Nagy MA, Sabatini BL, Greenberg ME | 2018 | Single-cell analysis of experience-dependent transcriptomic states in the mouse visual cortex | https://www.ncbi.nlm.nih.gov/geo/query/acc.cgi?acc=GSE102827 | Gene Expression Omnibus, GSE102827 |
| Tasic B, Menon V, Nguyen TN, Kim TK, Yao Z, Gray LT, Hawrylycz M, Koch C, Zeng H | 2016 | Adult mouse cortical cell taxonomy by single cell transcriptomics | https://www.ncbi.nlm.nih.gov/geo/query/acc.cgi?acc=GSE71585 | Gene Expression Omnibus, GSE71585 |
| Baron M, Veres A, Wolock SL, Faust AL, Gaujoux R, Vetere A, Ryu JH, Wagner BK, Shen-Orr SS, Klein AM, Melton DA, Yanai I | 2016 | A Single-Cell Transcriptomic Map of the Human and Mouse Pancreas Reveals Inter- and Intra-cell Population Structure | https://www.ncbi.nlm.nih.gov/geo/query/acc.cgi?acc=GSE50244 | Gene Expression Omnibus, GSE50244 |

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
