## [Decision Letter]

Thank you for submitting your article "Identifying Gene Expression Programs of Cell-type Identity and Cellular Activity with Single-Cell RNA-Seq" for consideration by *eLife*. Your article has been reviewed by three peer reviewers, and the evaluation has been overseen by a Reviewing Editor and Naama Barkai as the Senior Editor. The following individuals involved in review of your submission have agreed to reveal their identity: Elisabetta Mereu (Reviewer #1); Berthold Göttgens (Reviewer #3).

The reviewers have discussed the reviews with one another and the Reviewing Editor has drafted this decision to help you prepare a revised submission.

Summary:

The non-negative matrix factorization method for the analysis of RNA-seq data presented in this paper, consensus-NMF (cNMF), addresses the identification cell types in the context of other non-cell type specific activities. and gene expression activity. A strength of the study is that results are not only compared with other methods, but also applied to both synthetic and real datasets.

Essential revisions:

A number of key elements of the algorithm are not described in sufficient detail to make it clear and reproducible, namely:

1) Clarify the comparison with alternative clustering approaches applied to the same problem, including the analysis of how it differs of other methods in the identification of genes that are part of cell activity programs, given the definition of cluster-specific marker is often done by a differentially expression analysis.

2) Provide an accurate description of the math of the method. Add equations to explain how to achieve the results and explain how parameters were selected.

2a) There is a large number of parameters used in simulation, data processing, running the method and methods comparison, many of which are not adequately explained or sufficiently justified (including but not limited to: uniform proportions of cell types in simulation, scaling non-log-transformed expression value in pre-processing, the choice of the number nearest neighbour for filtering outlier components, using 14 PCs for calculating cell distance for louvain clustering).

2b) The selection parameter K is non-trivial. It is a balance between stability and error. Currently, the choice of the number of factors, K, when applying the method to the simulated and the real datasets were not convincing. For example, k=13 seems more appropriate than k=14 for the simulated data judging from Figure 2—figure supplement 2A It's not clear which k should be chosen from Figure 3—figure supplement 1A. Showing how derived GEPs look under different choices of K would be essential. In addition, it is also unfair to compare with other unsupervised methods (such as PCA) if the K cannot be automatically selected. The authors can suggest an automatic solution to the K.

3) Pay more attention to performance according to the balance between cell types and samples.

3a) Because batch effects strongly affect RNAseq analysis, it is crucial the identification of a good set of highly variable genes before clustering for example or the application of a batch correction. How authors can be sure that one or more GEPs that come out from cNMF are not reflecting batch effects? How the method controls batch effects?

3b) There is not guarantee that one can find factors correlating with cell identity and activity in every dataset. It would be highly desirable to test the method in datasets with moderate batch effects and also in other tissues as currently tested real datasets are all in brain/neuronal tissues. It would be informative to show the variance explained by the overall model and each factor in both simulated and other real datasets preferably with a larger K to put the activity GEPs into perspective.

---

## [Author Response]

Essential revisions:A number of key elements of the algorithm are not described in sufficient detail to make it clear and reproducible, namely:1) Clarify the comparison with alternative clustering approaches applied to the same problem, including the analysis of how it differs of other methods in the identification of genes that are part of cell activity programs, given the definition of cluster-specific marker is often done by a differentially expression analysis.

We have significantly expanded our discussion of how cNMF allows marker gene identification and how this compares to differential expression approaches that follow hard-clustering. We describe this in a new section in our revised “Materials and methods” titled “Identification of marker genes”, and further explain our reasoning below.

State-of-the-art software packages for single-cell RNA seq analysis such as SEURAT (Butler et al., 2018) and Scanpy (Wolf, Angerer, and Theis, 2018) compare gene expression of cells in a cluster to the expression of all other cells not in the cluster (or to cells in another cluster) using group-comparison hypothesis tests such as unpaired T-test and Wilcoxon Ranksum test. Given that cNMF does not assign cells to discrete groups, we cannot use a standard differential expression approach for identifying marker genes, and therefore adapted a solution to our setting.

We identify marker genes by fitting multivariate linear regression models of (z-scored) gene expression profiles against GEP usage. This approach generalizes the commonly-used T-test to settings where cells have continuous weights for each GEP, rather than binary assignments. This is now described in explicit mathematical notation as per essential revision #2 below.

Our decision to identify marker genes with multivariate linear regression as specified above is motivated by several key considerations:

1) This framework allows us to compare hard and soft-clustering approaches within the same testing framework, e.g., in Figure 2E.

2) Regressing z-score normalized expression profiles rather than raw expression values makes the coefficients comparable across all genes, even between genes that are expressed at very different magnitudes. The coefficients all represent changes relative to the average expression across all cells.

3) While in principle we could detect marker genes with separate tests for each GEP (e.g. by identifying genes with high Pearson correlation between the GEP usage and gene expression), this could suffer from confounding if GEPs are not independent. For example, if a gene is a marker of a cell-type that frequently expresses an activity program, there could be positive correlation between the activity usage and the gene’s expression, even if the gene isn’t directly associated with the activity. Including both the identity and activity GEP as covariates avoids this confounding and strengthens our testing approach.

4) The regression coefficients are unbiased estimators of the association between GEP usage and gene expression, even though gene expression data is not normally distributed. We do not use the P-values of the regressions at any point in our analysis as those can be inaccurate due to non-normality.

2) Provide an accurate description of the math of the method. Add equations to explain how to achieve the results and explain how parameters were selected.

In the revised manuscript, we have formalized the presentation of the method using detailed mathematical notation intertwined with the text description of the method. We agree that this will increase the precision of the method description and will provide a clearer description for mathematically inclined readers. In our response to 2a, we describe the changes we made to clarify our parameter selection approach and, where applicable, we have added corresponding equations in the “Materials and methods”.

2a) There is a large number of parameters used in simulation, data processing, running the method and methods comparison, many of which are not adequately explained or sufficiently justified (including but not limited to: uniform proportions of cell types in simulation, scaling non-log-transformed expression value in pre-processing, the choice of the number nearest neighbour for filtering outlier components, using 14 PCs for calculating cell distance for louvain clustering).

We have added clarifying text, primarily in the relevant sections of the “Materials and methods”, explaining the motivation and procedures behind non-trivial technical decisions and parameter choices, including all those raised by the reviewer and others. In the sections below, we expand on these additions and describe where they can be found in the main text.

Uniform proportions of cell-types in simulations:

We acknowledge that our simulation of cell-types at uniform proportions is a simplification of the biological reality, where cell-type frequencies can vary over multiple orders of magnitude. We did so to simplify our benchmarking analysis, as it allows us to treat all identity programs as replicates of each other for evaluating inference accuracy. If we had alternatively simulated cell-types with variable proportions, this simplification would not have been possible because GEPs of rare cell-types would (everything else equal) be harder to infer than those of common cell-types.

It is important to ensure that the simplification of simulating cell-types at uniform proportions does not change our overall conclusions about the applicability or comparative performance of the methods. Therefore, in the revised text, we describe additional simulations where cell-type proportions matched those of a representative real biological dataset (the Hrvatin et al. visual cortex data), where cell-type proportions ranged from 0.6% to 21.5% with four cell-types below 1%. This analysis is described in the last paragraph of the simulation section of the Results (Figure 2—figure supplement 8). We found that cNMF, cICA, and Louvain clustering all failed to identify two or more rare cell types when the effect-size parameters (which control how distinct the GEPs are from each other) were kept the same as in the primary simulation. However, both cNMF and cICA identified all of the GEPs when the effect-size was increased from 1.0 to 2.0 for differential expression mean. We did not test cLDA because it is prohibitively slow to run.

Our findings are consistent with an intuitive explanation: rarer cell types will be harder to identify unless their identity GEPs are more distinct from those of other cell types in the data. cNMF performed at least as well (or better) relative to the other methods in these simulations, compared to the simulations where cell type proportions were uniform. It identified 13/15 identity GEPs in the small effect-size simulation, compared to 12/15 for cICA and Louvain clustering, and better deconvoluted the activity GEP than the other methods in the large effect-size simulation. Thus, we conclude that the simplification of using uniform cell-type proportions does not change the conclusions of our analysis.

Scaling non log-transformed expression values in pre-processing

We have added additional text in the preprocessing section of the “Materials and methods” and a citation for a recent preprint by Townes et al., 20 to justify our decision to variance-normalize non log-transformed expression data. We briefly expand on this decision below.

It is helpful to transform gene expression data to move genes on different expression magnitudes to the same scale so that a few very highly expressed genes do not dominate the signal. This is typically accomplished by log-transforming the data, but the same function can be accomplished by normalizing the data so that all genes have a variance of 1.

Log-transforming gene-expression data is commonly used, but has several drawbacks. First, it requires the addition of a pseudocount to the data, the precise value of which is arbitrary but yet can have a large impact on the clustering. Second, for genes whose expression varies over multiple orders of magnitude, log-transformation makes a fold-change of 10 to 100 and 0.1 to 1 equivalent in absolute terms. Given the resolution of current single-cell data, we feel that this is undesirable because the change from 0.1 to 1 can easily be due to noise while we would have much higher confidence in the 10 to 100 fold change.

By variance normalizing non log-transformed data, we still ensure that genes on different expression-level scales contribute equivalent variance to the cNMF model. However, this approach does not require the addition of an arbitrary pseudo-count nor any transformation of the shape of the distribution of gene expression.

To further justify this decision, we now cite a recent preprint by Townes et al. that empirically demonstrates the downsides of log transforming scRNA-Seq data. It also describes an alternative pre-processing approach, termed multinomial deviance residuals, which is the equivalent of Z-scoring normally-distributed data under a multinomial probabilistic model. They demonstrate that their preprocessing performs better than log-transforming data for an unsupervised clustering task (Townes et al., 2019). Variance normalization serves an analogous function to Z-scoring gene expression prior to PCA, but omits the mean-centering step (which would make the data negative and thus incompatible with NMF). This preprint supports our premise that variance normalization preprocessing approaches can outperform log-transformation for unsupervised analysis of scRNA-Seq data.

Using 14 PCs for calculating cell distances for Louvain clustering

We have clarified this logic in the “Comparison of cNMF with other methods” part of the “Materials and methods” section and further elaborate below.

Typically, scRNA-Seq data undergoes linear dimensionality reduction (Principal Component Analysis, PCA) and only a subset of the PCs are used for cell clustering. This speeds up computations by decreasing the dimensionality of the data, and also denoises the data, as PCs beyond a certain number are assumed to correspond to noise rather than true biological signal. In this case, because we have simulated the data based on a true latent dimensionality of 14, we can be confident that all PCs beyond 14 correspond to noise and that the optimal number of PCs for this task is not greater than 14. In real biological cases without ground truth, the number of PCs is often selected by choosing an elbow in the ‘Scree plot’ (or related methods) which shows the variance explained per PC (see e.g. Consortium et al., 2018; Ordovas-Montanes et al., 2018). We confirmed that the choice of 14 is supported by the elbow in the Scree plot, which we have added to Figure 2—figure supplement 3.

Using 200 nearest neighbors for Louvain clustering

In the first draft of the manuscript we used 15 nearest neighbors (the Scanpy default in the version we were using at the time) for Louvain clustering. In the revised manuscript, we have increased this number, in keeping with the idea that the number should be as large as possible (in order to minimize variance), but only up to the minimum number of cells you expect to belong to the smallest distinct population in the data. If the number of neighbors exceeds the size of the smallest population, the clustering will obscure that population by smoothing over cells of other populations as well. For the simulations, the smallest population in the data would be the 30% of cells of a distinct cell-type that express the activity program. This should be approximately 0.3*15000*(1/13)=346. In reality, one does not know the true sizes of the populations in the data and would instead choose a number corresponding to the a priori belief of what this smallest distinct population in the data might be. 200 is a reasonable choice for such analyses and performs well in practice on the simulated datasets to which we applied it. This logic is described in the “Comparison of cNMF with other methods” section of the “Materials and methods”.

Use of 2000 overdispersed genes for cNMF on all datasets

We added the following text to the relevant part of the ‘Materials and methods’ section to clarify the decision to use 2000 over-dispersed genes:

“We then selected the H most over-dispersed genes as determined by the v-score for input to cNMF. […] 2000 reflects a trade-off between including enough genes to detect subtle biological signals, and speeding up computation by not including too many extraneous genes.”

We further note that 2000 is a commonly used choice (e.g. Lake et al., 2018; Rodda et al., 2018; Knier et al., 2018) and is the default for the SEURAT version 3 FindVariableFeatures as of April 03, 2019).

Number of nearest neighbors for outlier filtering

We have clarified this point in the “Consensus Non-negative Matrix Factorization” section of the “Materials and methods” and summarize the discussion below.

We frame the decision of choosing the number of nearest neighbors for outlier filtering as one of choosing a fraction of simulation replicates (ρ) that must yield a component approximately matching a GEP in order for that GEP to be retained by cNMF. The K nearest neighbor distance threshold (τ) determines what constitutes how similar the components must be to constitute approximately matching.

While we originally set ρ=0.3 for all analyses of real data in this manuscript and ρ=0.35 for all simulations, for consistency, we went back and re-ran all simulation analyses with ρ=0.3. This did not change the results so now we use ρ=0.3 for all analyses.

A choice of ρ=0.3 implies that we would consider components that arise approximately in 30% of simulation replicates to constitute consensus GEPs. The fact that we can use the same parameter for 4 very different real datasets from 3 sequencing technologies and several distinct simulations demonstrates that this is an appropriate default setting for the method. In general, we recommend consulting the consensus clustergram for the data (shown in Figure 2—figure supplement 3, Figure 3—figure supplement 1, and Figure 4—figure supplement 1) to determine which correlation blocks seem robust, and selecting the number of nearest neighbors to be bigger than the largest correlation block that should be filtered out (e.g. because it appears as a small sub-block of a larger correlation block). However, we believe that, in practice, ρ=0.3 is an appropriate default choice for most applications.

2b) The selection parameter K is non-trivial. It is a balance between stability and error. Currently, the choice of the number of factors, K, when applying the method to the simulated and the real datasets were not convincing. For example, k=13 seems more appropriate than k=14 for the simulated data judging from Figure 2—figure supplement 2A. It's not clear which k should be chosen from Figure 3—figure supplement 1A. Showing how derived GEPs look under different choices of K would be essential. In addition, it is also unfair to compare with other unsupervised methods (such as PCA) if the K cannot be automatically selected. The authors can suggest an automatic solution to the K.

Thank you for this comment which led us to make the following key additions to the revised manuscript:

1) Expanding the discussion of the considerations for choosing K in the “Choosing the number of components” section of the “Materials and methods”;

2) Adding a robustness analysis demonstrating that our findings are consistent across multiple choices of K (Figure 5);

3) Using scree plots as an additional graphical aid in the selection of K (Figure 2—figure supplement 3, Figure 3—figure supplement 1, and Figure 4—figure supplement 1).

We discuss each of these additions below. However, first we must respectfully disagree with the reviewer’s claim that it is unfair to compare with other unsupervised methods such as PCA if the K cannot be automatically selected. Even for PCA, K selection remains a decision that must be weighed carefully, and although tools such as scree plots can guide its selection, no general approach can be trusted to be applied blindly. The challenge of selecting the number of PCs for scRNA-Seq analysis is well described in the SEURAT guided clustering tutorial (https://satijalab.org/seurat/pbmc3k_tutorial.html, Compiled: April 5, 2019) which provides 3 different approaches to this task:

“Identifying the true dimensionality of a dataset – can be challenging/uncertain for the user. […] We advise users to err on the higher side when choosing this parameter. For example, performing downstream analyses with only 5 PCs does significantly and adversely affect results.”

All 3 discussed approaches, in this tutorial for a widely used scRNA-Seq analysis package, require an element of human input in the decision.

We further believe that finding an optimal K may be a biologically ill-defined problem. The intrinsic dimensionality of gene expression is potentially very high, with a long tail of dimensions that explain an increasingly small, but non-negligible, amount of the data. The optimal choice of K may reflect a trade-off between sensitivity to detect true dimensions, and tolerance for identification of spurious dimensions, which should be specified by the user based on the downstream application. Furthermore, biological systems frequently have a hierarchical organization – e.g. in immunology, there are high-level myeloid and lymphoid lineages, which encompass specific cell-types (e.g. T-cells) which in turn encompass sub-cell-types (e.g. helper T-cells) and sub-sub-cell-types (e.g. Th1 or Th17 cells), etc. While there may eventually be a bottom of this hierarchy, in practice, our datasets are too small and lack sufficient resolution to discern all of it. In the meantime, there are potentially multiple appropriate choices of K depending on the level in the hierarchy at which you stop. In this context, the most resolved choice of K should depend on what can be robustly inferred from the data before too much noise begins to contaminate the results. Ultimately, this decision requires human input and reasoning.

The above thought process as well as the previously published procedure in the context of identifying mutational signatures(Alexandrov et al., 2013), motivated the graphical aid we described in the manuscript. However, based on this helpful reviewer comment, it is now clear that we omitted a key aspect of the overall decision process which we have tried to clarify in the “Choosing the number of components K” section of the revised “Materials and methods”. The stability vs. error plots shown in figure supplements are just a starting point and a guide in the K selection process, and it is not the case that one should necessarily choose the maximum stability solution. In analyzing the data, we considered multiple Ks in the vicinity of the maximum stability solution, and chose a solution based on the biological interpretability of the factors we found. To our knowledge, this is advisable for the choice of PCs (as is stated in the paragraph from the SEURAT tutorial quoted above) as well as for the resolution parameter in Louvain community detection (which determines the number of clusters).

Given this ambiguity in the choice of K, we agree with the reviewer that it is very important to show that varying the choice of K will not significantly change our conclusions. Therefore, we have added robustness analyses of the primary datasets analyzed in the manuscript, comparing what we find when a range of K values are chosen to the results presented in the manuscript. We have added Figure 5 which shows that, empirically, the same core set of GEPs are found for a range of K values with essentially a single new GEP being discerned with each step in K. For each step below the selected K, approximately a single GEP is lost, but for choices above the selected K, components approximately matching the original K programs (I.e. with Pearson correlation >.7) are typically found.

To further address the reviewer’s concern about the utility of the stability-vs.-error plots, we now include Scree plots as figure supplements to aid in the selection of K. The idea is that choosing the number of principal components and choosing the number of NMF components are very related problems, as they can both be framed as estimating the rank for a low-dimensional representation of the data. The Eckart Young Mirsky theorem implies that PCA will provide the matrix factorization with minimum Frobenius reconstruction error for any choice of K (Eckart and Young, 1936), and we also use the Frobenius error in our NMF model. Because principal components are orthogonal to each other while NMF components in general are not, and NMF usages can never be negative, K principal components will always span a larger sub-space than K NMF components. This suggests that the optimal number of NMF components will likely not be smaller than the optimal number of PCs. We now include scree plots in the main figure supplements showing the proportion of variance explained per principal component and indicating where the choice of K used in the analysis falls. In each case, it appears at a reasonable “elbow” on the plot. Although this still isn’t an automatic choice of K, we believe that it is the current standard in the field.

3) Pay more attention to performance according to the balance between cell types and samples.

To address this reviewer comment and the sub-comments 3a and 3, we have added a supplementary note titled “Analysis of between-sample variability**”** to the revised manuscript. In this note, we discuss the impact of variability between samples within the results presented in the main text and describe the analysis of a new dataset we analyzed where batch effect was a significant concern (described further in the response to essential revision 3b). This note clarifies how cNMF functions in the context of variation between samples and provides reassurance that GEPs discussed in the manuscript reflect real biological signal and are not technical artifacts due to batch-effect. See 3a and 3b below for a discussion of the new analysis that has been added to the supplemental note.

3a) Becaue batch effects strongly affect RNAseq analysis, it is crucial the identification of a good set of highly variable genes before clustering for example or the application of a batch correction. How authors can be sure that one or more GEPs that come out from cNMF are not reflecting batch effects? How the method controls batch effects?

cNMF does not directly control for batch effects. If there is signal reflecting batch-to-batch variation in the data, cNMF will incorporate that signal into one of more of the inferred GEPs. If batch-effect is a significant artifact of the data, we recommend applying one of the recently published batch-effect correction approaches (Haghverdi et al., 2018; Butler et al., 2018; Johnson, Li, and Rabinovic, 2007) prior to running cNMF. However, we note that batch effect correction risks removing real signal from the data and thus should be used with caution.

In this manuscript, we focused on 2 real datasets that each contained data aggregated from dozens of biological replicates. We don’t necessarily expect the relative compositions of GEPs to be identical across replicates. Quadrato et al., 2017 described significant variability between organoid replicates that were primarily attributable to organoids grown in some bioreactors being more differentiated than those grown in others. In their manuscript, they performed immunohistochemistry validation experiments to confirm that this reflected true biological differences between samples. The fact that we identify more differentiated cell-types in organoids from those bioreactors represents a batch-effect but also represents true biological signal that we would hope for cNMF to discover.

To explore between-sample variability, we have added a new supplementary figure to the revision that compares the GEP profile of each replicate (left panels of Figure 5—figure supplement 1), and the proportion of signal derived per replicate for each GEP (right panels) for the visual cortex and organoid datasets. The first approach lets us identify between-sample variation in the data, which could either be technical or biological in nature. The second approach allows us to identify GEPs that occur predominantly in one or two replicates and thus may not be reproducible within the experiment.

In the Hrvatin et al. data, the GEP patterns of the different mouse replicates were very homogeneous, except for the expected pattern that the depolarization-induced GEPs (ERP, LRP-S, and LRP-D) primarily occur in mice treated with the 1h and 4h stimuli respectively. Every GEP arose in multiple mouse replicates suggesting that they are reproducible.

By contrast, in the organoid data, there were several clusters of organoids that had distinct GEP profiles. The predominant pattern matched what was described in the Quadrato et al., 2017 manuscript where forebrain GEPs were predominantly represented in organoids from bioreactor 3, and organoids in bioreactor 4 had a greater representation of undifferentiated precursors (e.g. proliferative precursors and neuroepithelial cells). These differences between bioreactors and organoids represent validated biological differences and are unlikely to reflect technical artifacts. Again, every GEP arose in multiple organoid replicates with none deriving more than 45% of their total usage from a single replicate. Each GEP was the maximum contributor for ≥1 cell in at least 6 distinct organoid replicates. This provides support that the inferred GEPs represent reproducible biologically meaningful signals within the dataset.

3b) There is not guarantee that one can find factors correlating with cell identity and activity in every dataset. It would be highly desirable to test the method in datasets with moderate batch effects and also in other tissues as currently tested real datasets are all in brain/neuronal tissues. It would be informative to show the variance explained by the overall model and each factor in both simulated and other real datasets preferably with a larger K to put the activity GEPs into perspective.

To simultaneously address the suggestions to analyze both non-neuronal tissue and data containing moderate batch effects, we re-analyzed a published dataset of human pancreatic islets with noted variability between the four donor-derived samples that constituted the data (Baron et al., 2016).

This data does not have a larger K than the organoid dataset we analyzed previously (which had K=31). However, we do not recommend using cNMF for datasets with K larger than that. For data containing many divergent cell-types, there is less resolution to pick up more subtle differences within cell-types in the data. For example, in our analysis of the Hrvatin et al. visual cortex data, we ran cNMF only on the neuronal cells which were defined based on the published clustering of the data. We excluded the many non-neuronal cells that made up a significant portion of the Discussion in the original publication. This is because our goal was to identify activity programs, which can be relatively more subtle than cell-type programs. The activity programs would have been swamped out by the much greater variation that exists between a neuron and glia, or a neuron and an endothelial cell, had we kept glia and endothelial cells in the data.

For datasets containing many highly divergent cell-types, it can be beneficial to cluster the data first and run cNMF separately on the distinct clusters. This is analogous to the 2-stage clustering performed in the Hrvatin et al. publication where they first clustered to identify broad cell-lineages like neuronal and glial, then sub-clustered those clusters to identify specific cell-types within those lineages. We have now highlighted the benefit of analyzing clusters of related cell populations in the revised manuscript where we introduce the visual cortex dataset.

In our analysis of the pancreatic islet data, we found multiple identity programs for many cell-type clusters that corresponded to “donor of origin” (Figure 5—figure supplement 2). Importantly, in this analysis, cNMF did not learn a single GEP for each donor but rather identified multiple hybrid identity-donor GEPs corresponding to individual cell-types derived from distinct sets of donors. This is likely due to the fact that the batch effect had different impacts on gene expression in different cell-types. Therefore, no single shared “batch-effect” GEP could capture the impact of batch on each cell-type. This is illustrated in “Author response image 1” which shows a heatmap of average expression levels of many genes that were differentially expressed across donors in at least one of 3 cell-type clusters.

**Author response image 1. respfig1:** Genes that are differentially expressed between donors in Baron et al., 2016. We performed differential expression analysis comparing gene expression between cells derived from each donor using ordinary least squares regression of donor dummy variables on log10 TPM expression values. We did this separately for cells assigned to the acinar, α, and β cells clusters from Baron et al., 2016. Shown above are the regression coefficients for the 20 genes with the most significant batch-effect for each cell-type, as ranked by the F-test P-value. While some signals are common across cell-types, others show marked variation between cell-types.

These results illustrate the reviewer’s point that it is important to consider batch effects in the analysis of single-cell RNA-Seq data. They suggest that in the context of moderate batch-effect, cNMF will likely not learn distinct technical components corresponding to batch alone but will instead learn hybrid batch-identity GEPs which must be interpreted correctly. Batch effect correction prior to cNMF can help address this challenge.

As was suggested in this reviewer comment, we have also computed the variance explained by cNMF for the simulated dataset and the real datasets analyzed in the manuscript (Author response image 2).

**Author response image 2. respfig2:** Variance explained by cNMF and PCA. Variance explained by the cNMF solutions presented in the manuscript and Principal Components with the same K as was used for cNMF. Explained variance is calculated for the input matrix used for cNMF (2000 most over-dispersed genes, variance normalized).

We compare this to the overall variance explained by the same number of principal components, which, as a consequence of the Eckart Young Mirsky theorem cited previously, will necessarily be greater than the variance explained by cNMF. We note that the proportion of variance explained by both cNMF and PCA is low: less than 40% for the datasets analyzed. This is a consequence of the extensive transcriptional noise in single-cell RNA-Seq data. We note that unlike for PCA, the cNMF components are not orthogonal to each other. As a result, the variance explained by all of the NMF components together is not equal to the sum of the variance explained by each NMF component individually. Treating each NMF component as an independent predictor and computing the reconstructed matrix, we find that the residual sum of squares is greater than the total sum of squares, making the explained variance negative. As a result, it doesn’t make sense to compute the variance explained by each factor for NMF like one would for PCA.